# Development, validation, and pilot MRI safety study of a high-resolution, open source, whole body pediatric numerical simulation model

**Hongbae Jeong**[1,2], **Georgios Ntolkeras**[1,3], **Michel Alhilani**[3,4], **Seyed Reza Atefi**[1,2], **Lilla Zöllei**[1,2], **Kyoko Fujimoto**[5], **Ali Pourvaziri**[2], **Michael H. Lev**[2], **P. Ellen Grant**[3], **Giorgio Bonmassar**[1,2] *

**1** Athinoula A. Martinos Center for Biomedical Imaging, Massachusetts General Hospital, Harvard Medical School, Boston, MA, United States of America, **2** Department of Radiology, Massachusetts General Hospital, Harvard Medical School, Boston, MA, United States of America, **3** Fetal-Neonatal Neuroimaging and Developmental Science Center, Boston Children's Hospital, Harvard Medical School, Boston, MA, United States of America, **4** Department of Medicine, Charing Cross Hospital, Imperial College Healthcare NHS Trust, London, United Kingdom, **5** Center for Devices and Radiological Health, U. S. Food and Drug Administration, Silver Spring, MD, United States of America

* Giorgio.bonmassar@mgh.harvard.edu

**Data Availability Statement:** The proposed voxel model will be available on the Analogue Brain Imaging Laboratory (ABILAB) at the Athinoula A.

## Abstract

Numerical body models of children are used for designing medical devices, including but not limited to optical imaging, ultrasound, CT, EEG/MEG, and MRI. These models are used in many clinical and neuroscience research applications, such as radiation safety dosimetric studies and source localization. Although several such adult models have been reported, there are few reports of full-body pediatric models, and those described have several limitations. Some, for example, are either morphed from older children or do not have detailed segmentations. Here, we introduce a 29-month-old male whole-body native numerical model, "MARTIN", that includes 28 head and 86 body tissue compartments, segmented directly from the high spatial resolution MRI and CT images. An advanced auto-segmentation tool was used for the deep-brain structures, whereas 3D Slicer was used to segment the non-brain structures and to refine the segmentation for all of the tissue compartments. Our MARTIN model was developed and validated using three separate approaches, through an iterative process, as follows. First, the calculated volumes, weights, and dimensions of selected structures were adjusted and confirmed to be within 6% of the literature values for the 2-3-year-old age-range. Second, all structural segmentations were adjusted and confirmed by two experienced, sub-specialty certified neuro-radiologists, also through an interactive process. Third, an additional validation was performed with a Bloch simulator to create synthetic MR image from our MARTIN model and compare the image contrast of the resulting synthetic image with that of the original MRI data; this resulted in a "structural resemblance" index of 0.97. Finally, we used our model to perform pilot MRI safety simulations of an Active Implantable Medical Device (AIMD) using a commercially available software platform (Sim4Life), incorporating the latest International Standards Organization guidelines. This model will be made available on the Athinoula A. Martinos Center for Biomedical Imaging website.

Martinos Center for Biomedical Imaging website after publication.

**Funding:** This work was supported by the NIH/ NIBB grant R01EB024343. The funders had no role in study design, data collection and analysis, decision to publish, or preparation of the manuscript.

**Competing interests:** N/A

## 1. Introduction

Computational modeling studies of the human body have been widely used in government, industry, and academia. In the medical device market, the safety and effectiveness of medical devices play an important role, and modeling can support the development from the design stage to the final marketing. The areas of computational modeling with human body are wide: fluid dynamics [1], electromagnetics [2], optics [3], ultrasound [4], thermodynamics [5], and mechanics [6, 7]. Computational modeling with virtual humans is helpful in studying the interaction of complex biological problems *in silico* [7], for source localization [8, 9], radio-frequency (RF) and specific absorption rate (SAR) exposure [10], and neurostimulation [11–13]. The accurate anatomical representation of human numerical models has become an integral part of many state-of-the-art safety studies, such as computed tomography (CT) dosimetry [14] and in MRI RF exposure [15–17]. The Visible Photographic Man (VIP-MAN) is a well-known, image-based whole-body model that was introduced by the National Library of Medicine's Visual Human Project [18, 19]. The cadaver of a 38-year-old male was imaged with four data modalities; X-ray, CT, MRI, and cryosection color photography, and the data were segmented into a 3D whole-body voxel model. More recently, the Virtual Family was introduced with various ages ranging between eight weeks and 80 years by IT'IS [20, 21]. The segmentation of tissue compartments was done using medical images from whole-body in-vivo subjects. Numerical simulations of the human body are realistic when the high-resolution segmented tissues are anatomically accurate, and when using the correct dielectric and thermal properties [15, 22, 23]. By far, one of the most anatomically accurate head and neck models is MIDA [16], which is an open-source model that includes the 115 structures with individual nerve tracts and deep brain structures, and it was used for estimating the MR signal intensities [24], neuronal stimulation [25, 26], and the assessment of electromagnetic (EM) field distribution in the head [27]. Furthermore, the higher level of detail in the high-resolution model led to a better estimation of the SAR exposure in the vertebrae, where the layer of cortical/cancellous bones can clearly be distinguished; by comparison, a low-resolution model included only one type of bone in the vertebrae [21]. The most anatomically accurate child model to date is Roberta [21], which is a model of a 5-year-old female. However, all the 3-year-old models currently available are not anatomically accurate (see Table 1 and S1 File for the complete survey of the existing pediatric models). For instance, Nina is a 3-year-old female model, and it was morphed from Roberta, which resulted in the anatomical inaccuracies due to the anisotropic growth of different tissues since the body tissues do not grow proportionally during childhood. For example, the heart of a 6-year-old child weighs 1.59 times more than that of 3 years old child while the brain weights only 1.09 times more, which can represent a 68.6% relative difference between the growth of two different tissues [28].

In order to address these limitations, we introduce an open-source 29-month-old whole-body voxel model (MARTIN). The segmentation was done based on in-vivo MRI and CT data, which allow accurate tissue segmentation for a 29-month-old child. Unlike some other models, all the details of brain structures were segmented without morphing. The quality of segmentation was quantified using metrics (i.e., Dice similarity coefficient and Hausdorff distance), volumetric information, and by numerically comparing the Bloch simulation estimations with MARTIN to the original MRI data.

The investigation of RF safety in children is of interest since RF exposure in pediatric head tissues differs from that of adults [36, 37]. We chose a vagus nerve stimulation (VNS) implant as our pilot model for simulation testing because this is a well-studied, important application in the literature [38, 39]. Indeed, it was reported that more than 30,000 children had used a VNS [40], and 60 to 90% seizure reduction has been achieved in four out of six patients [39].

**Table 1. List of pediatric whole-body models currently available.**

| Model Name | Charlie | Nina | Roberta | Korean Child model ETRI | UF Family of reference hybrid phantoms | GSF Family (Baby,Child) | Pediatric xCAT phantoms | Chinese family phantom USTC |
|---|---|---|---|---|---|---|---|---|
| Year of model released | 2014 | 2014 | 2014 | 2009 | 2010 | 2003 | 2015 | 2017 |
| Imaging modalities used | GSF family, Baby [29] | Roberta [21] | MRI | MRI | CT | CT | PET-CT | RPI mesh phantom [30] |
| Age imaging scans used for segmentation | Eight-week-old | 3-year-old | 5-year-old | 7-year-old | Newborn, 1-,5- and 10-year- old | Eight-week-old and 7-year-old | Newborn,1-,5-,10-,15-year-old | 5-, 10-,15-year-old |
| Number of tissue varieties | 45 tissues | 75 tissues | 78 tissues | 44 tissues | 47 tissues | 41 tissues | 64 tissues | 50 tissues |
| Segmented both white- and grey-matter | Yes | Yes | Yes | Yes | No | No | No | No |
| Morphing or adjustments used | No | Yes | No | Yes | Yes | Yes | Yes | Yes Database: 50th percentile population |
| Method of validation | Anatomical knowledge | Anatomical knowledge | Anatomical knowledge | Comparison with the physiques measurement of reference models | Comparison with the physiques measurement of reference models | Comparison with the physiques measurement of reference models | Comparison with the physiques measurement of reference models | Comparison with the physiques measurement of reference models |
| Is the model freely accessible to the research community | No | No | No | No | Yes, upon request | Yes, upon request | No | Information not available |
| References/ company | Petoussi-Henss et al., 2002 [29] /IT'IS [21] | IT'IS [21] | IT'IS [21] | Lee et al., 2009 [31] | Lee et al., 2010 [32] | Petoussi-Henss et al. 2002 [29] / Helmholtz Zentrum München | Segars et al., 2015 [33, 34] | Pi et al., 2017 [35] |

Although VNS was approved by the US Food and Drug Administration (FDA) for the treatment of pediatric epilepsy, no citations have been available through PubMed regarding the MR safety of the VNS device in children. MRI, however, is a routine diagnostic modality that may be required in younger patients with VNS implants, for follow up or identifying consequent comorbidities. As per Shellock et al. [41], important considerations with VNS implants include excessive heating in MRI scanners using body RF transmit coils. Thus, the VNS Therapy (Cyberonics/LivaNova, Houston, TX) implantable device for treating seizures comes with a warning of potential significant heating with VNS therapy when the RF body transmit coil is used (a 30°C increase or higher during head or MRI scanning) [42]. Thus, the current VNS product label lists several MRI restrictions to avoid tissue heating by limiting head-averaged SAR and spatial gradient field, as well as avoiding the use of the RF body transmit coil in patients with VNS implants. Several clinical studies conducted at 1.5 T and 3T have shown that adults [43, 44] and younger subjects with ages between 5 and 12 [45] can be scanned in MRI under controlled condition using a transmit/receive head coil without any report of heating, discomfort, or any other unusual sensation. The numerical simulations of MRI RF safety of pediatric subject with VNS are not presented in literature yet. In our simulation, the VNS lead was connected to an implantable pulse generator (IPG) and placed inside "MARTIN's" chest with neurologist guidance. The electromagnetic field distribution changes in the simulated child, with and without a VNS implant, were assessed using a head transmit coil at 1.5T, as per similar labeling for adults [42].

In summary, we have introduced MARTIN, a 29-month-old boy whole-body model, which includes 86 segmented tissues, segmented directly from MRI and CT images of a 29-month-old child, and was validated using three separate approaches. We show an example of RF safety simulations with MARTIN by preliminarily studying the case of a vagus nerve stimulation (VNS) implant in a 1.5 Tesla MRI.

## 2. Materials and methods

### The numerical model

**Subject and data acquisition.** A 29-month-old male (height: 86.1 cm, weight: 13 kg) was selected based on adequate image quality and lack of anatomical abnormalities in the CT and the MRI images (**Fig 1**) as well as the availability of multiple sequences that would facilitate the segmentation process (**Fig 2**). Subjective image quality was assessed by two experienced, sub-specialty certified neuro-radiologists with over 20 years of experience, P.E.G. and M.H.L., and images with adequate diagnostic quality, and without abnormal anatomy were selected to represent a healthy 29-month-old child. Images were retrieved from the picture archiving and communication system (PACS) database at Boston Children's Hospital. The data were acquired under IRB written approval (The Boston Children's Hospital Institutional Review Board and The Partners Human Research Committee) and in compliance with the health insurance portability and accountability act (HIPAA). The study protocol received approval by the Boston Children's Hospital (BCH) Institutional Review Board, which waived the need for written informed consent due to the study's retrospective nature.

The MRI data included both T1 and T2 weighted sequences were acquired with a SIEMENS TRIOTIM 3T scanner (SIEMENS Healthineers, Erlangen, Germany) from the top of the skull through the toes (**Table 2**). The CT scans were acquired on a SIEMENS SOMATOM DEFINITION FLASH (SIEMENS Healthineers, Erlangen, Germany) that covered the partial-head and neck (**Table 2**).

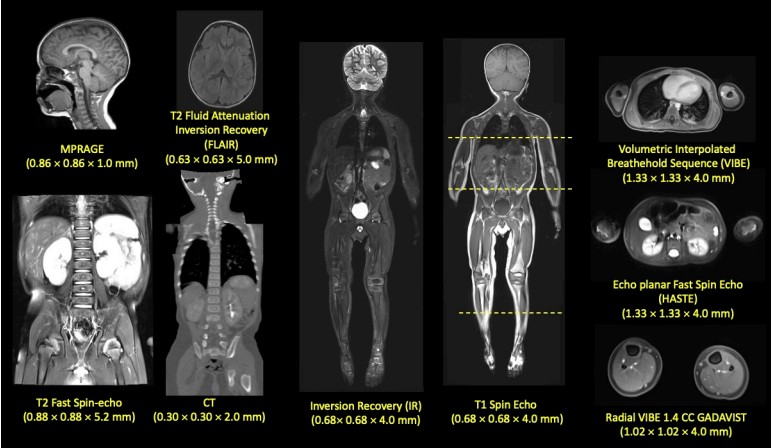

**Fig 1. Structural MRI and CT scans used for segmentation.** Several sequences with different resolution and contrast were used for the segmentation of different tissues. For example, MPRAGE, a T1 weighted image, was useful for the segmentation of brain structures while T2 Flair was used for the segmentation of the arteries and veins of the brain and Radial VIBE with a gadolinium-based contrast was used for the segmentation for the vessels of the lower extremities. CT was used for the segmentation of the cortical bone of the pelvis and the core, as well as the base of the skull. Inversion Recovery was useful for the segmentation of the CSF and the Vitreous body of the eyes. HASTE was initially for segmenting the kidneys and the CSF of the spinal cord as well. T2 Fast Spin Echo offered a good outline of the anatomy of the intervertebral disc. All the different tissues were finally referred to the whole body T1 space, while T1 was used for their manual refinement.

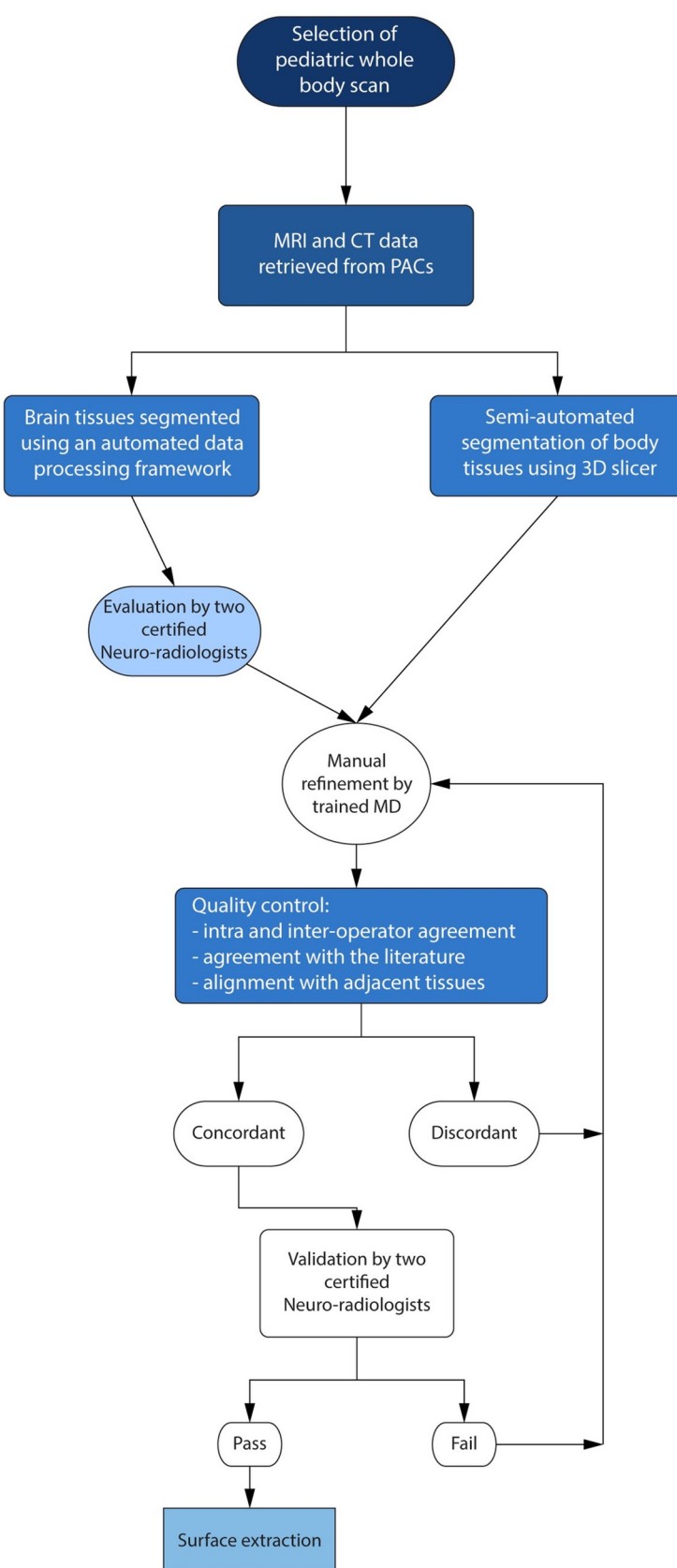

**Fig 2. Flow chart–process of the segmentation.** The first step was to select a male patient with available whole-body sequences without major deformities as well as multiple sequences that would facilitate the segmentation process. We then pulled all the available MRI and CT scans from the database of the Boston Children's Hospital. The meticulous process of the segmentation of the different tissues of the body was slightly differentiated for the brain and the non-brain tissues. For the segmentation of the brain structures, an automated infant-specific data processing framework was used, and the result was reviewed by two subspecialized neuroradiologists. Non-brain tissues were segmented using different sequences by two MDs, and the volumes and the weights of the segmented tissues were compared with the values from the literature. When the agreement was achieved between the operators and the tissues were aligned with the surrounded segmented tissues, the two sub-specialty certified neuroradiologists confirmed the result following a "pass or fail" process through an interactive process. The tissues that were scored with a "pass" were then finalized. Given that gradually more and more tissues were added to the segmentation project, some of which were not available when the first of the tissues were segmented and finalized, all the tissues were put together in the reference T1 sequence, and the model was again confirmed by the two neuroradiologists.

**Data co-registration.** Co-registration between different sequences and modalities was done using an extension tool in 3D Slicer, an open-source software platform [46].

i. *Registration between MRI sequences.* Linear registration (six affine degrees of freedom as rotation in x, y, and z and translation in x, y, and z) was done between MRI sequences using the whole-body coronal T1 image as a reference volume. For the registration of the MPRAGE, the details of the brain structures and non-brain structures were extracted using the Brain extraction tool (BET) in FMRIB Software Library (FSL) [47] to improve the alignment of the brain.

ii. *Registration between CT and MRI.* First, the linear registration (six affine degrees of freedom as rotation in x, y, and z and translation in x, y, and z) was used to align the CT image into the reference MRI image volume. The linear registration provided a starting point for the non-linear registration between the reference MRI volume and CT volume. Second, the non-linear registration was applied, since the subject was CT imaged in a different posture from the original MRI acquired six months before. The non-linear registration was done using the 3D Slicer extension tool, Elastix [48]. The reference volume in MRI was first to cut into the equivalent volume as three parts of the CT volume and registered into the CT volume. The warping matrix was then inverted and applied to the CT volume for the registration into reference MRI volume. **Fig 3** shows the sagittal view of T1 MRI with the registered CT chest and neck.

**Segmentation process.** Images from a whole-body 3T MRI were used to segment the soft tissues, including the uncalcified part of the bones. CT images of the lower head, chest, and

**Table 2. MRI and CT sequence parameters that were used for segmentation.**

| MRI | Name of sequence | Resolution (mm) | TR (ms) / TE (ms)/ TI (ms) / FA (°) | FOV (mm) | NSA |
|---|---|---|---|---|---|
| | MPRAGE | 0.86 × 0.86 × 1.0 | 1130/2.22/800(TI)/9 | 256 × 256 | 1 |
| | T2 FLAIR | 0.63 × 0.63 × 5.0 | 9000/137/2500(TI)/150 | 320 × 288 | 1 |
| | T2 Fast Spin-Echo | 0.88 × 0.88 × 5.2 | 2000/84/120 | 320 × 168 | 4 |
| | Inversion Recovery | 0.68 × 0.68 × 4.0 | 5000/107/200(TI)/125 | 384 × 277 | 1 |
| | T1 Spin Echo | 0.68 × 0.68 × 4.0 | 769/9.4/75 | 384 × 307 | 1 |
| | Volumetric interpolated breath-hold examination (VIBE) | 1.33 × 1.33 × 4.0 | 3.77/1.71/12 | 256 × 256 | 1 |
| | Echo Planar Fast Spin Echo (HASTE) | 1.33 × 1.33 × 4.0 | 750/53/160 | 256 × 143 | 1 |
| | Radial VIBE 1.4 CC GADAVIST | 1.02 × 1.02 × 4.0 | 3.78/1.85/12 | 260 × 260 | 1 |
| CT | SOMATOM definition FLASH | Resolution (mm) | CT dose index | FOV (mm) | NSA |
| | CT of neck | 0.18 × 0.18 × 2.0 | 1.53 | 512 × 934 | 1 |
| | CT of chest | 0.29 × 0.29 × 2.0 | 2.30 | 512 × 567 | 1 |
| | CT of abdominal | 0.30 × 0.30 × 2.0 | 3.71 | 512 × 904 | 1 |

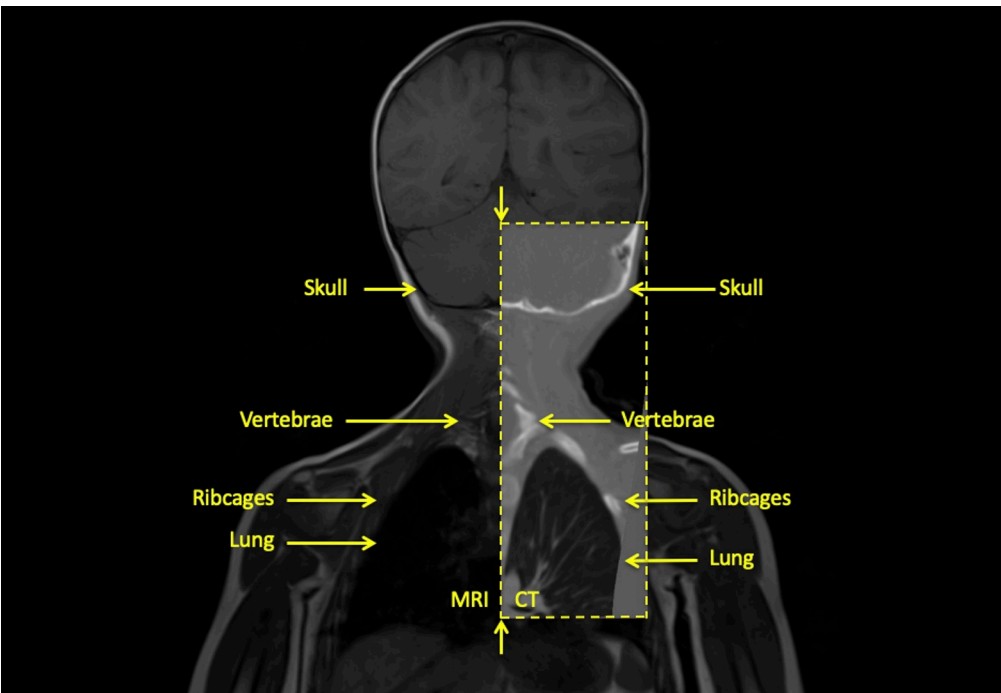

**Fig 3. MRI and CT registration.** Coronal view of the co-registered MRI and CT Flash neck. The contrast of skull and bone, e.g., vertebrae, ribcage, was higher in CT, while the brain, e.g., WM, GM, was enhanced in MRI.

abdomen were used to segment the calcified bones (**Fig 1**). Segmentation of the brain structures (**Fig 4**) was done on brain MRI using an automated infant-specific data processing framework developed by Zöllei *et al.* [49]. They designed a multi-atlas label fusion segmentation framework [50] where the ground-truth information from a labeled training data set could be used for the segmentation. Further details about their automated tools can be found in the literature [49]. Minor manual edits to the automatic segmentation were done to correct errors and improve the outcome of the automated segmentation process (**Fig 5**). In particular, the misalignment of the grey matter boundary was refined to obtain accurate CSF and meninges boundaries around the grey matter. All the tissues that were not segmented using the automated infant-specific data processing framework (Artery, Vein, claustrum in white matter, Septum pellucidum, CSF, meninges, Cranial Nerves-II, V, VIII, IX) were manually segmented (**Fig 6**).

Segmentation of the labels for non-brain anatomical regions of interest was performed using 3D Slicer [46]. Two additional tools were used to segment the upper skull that was not CT scanned [51, 52]. The upper skull was combined with the lower mandible that was segmented from CT volume and processed with manual refinement for higher accuracy by the two MD segmentors (G.N. and M.A.), who employed 3D Slicer for all the manual segmentation by tracing the boundaries of each tissue or organ on each slice section. IR sequence was used in the region of the spine (**Fig 7**). Since both the arteries and the veins were lacking any MRI contrast in the given sequences, especially in the head, a knowledge-based segmentation was done for the vessel segmentation (**Fig 8**). Finally, all the tissue labels were overlaid and inspected in the reference image space to avoid the intersection between segmentation labels and any empty space without a label (**Fig 9**). Remaining empty holes and unlabeled islands that were not detected during the segmentation were found and replaced with the label of the

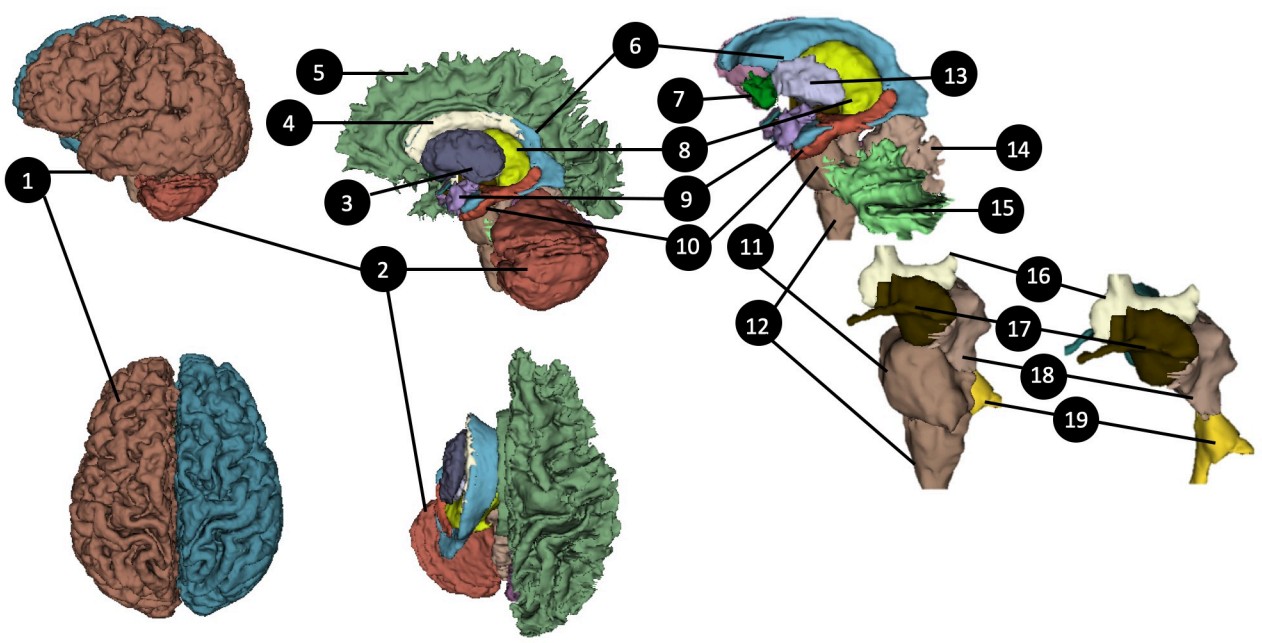

**Fig 4. 3D surfaces of the brain and its subcortical areas (created by automatic segmentation algorithm).** 1) Left-Cerebral-Cortex, 2) Left-Cerebellum-Cortex, 3) Left-Thalamus, 4) Left-Caudate, 5) Left-Cerebral-White-Matter, 6) Lateral ventricle, 7) Left-Accumbens-area, 8)Left-Putamen, 9) Left-Amygdala, 10) Left-Hippocampus, 11) Pons, 12) Medulla, 13) Left-Pallidum, 14) Vermis, 15) Left-Cerebellum-White-Matter, 16) 3rd-Ventricle, 17) Left-Ventral diencephalon (DC), 18) Midbrain, 19) 4th Ventricle. See S1 Table in S1 File for the list of tissues segmented by an automatic segmentation algorithm [42].

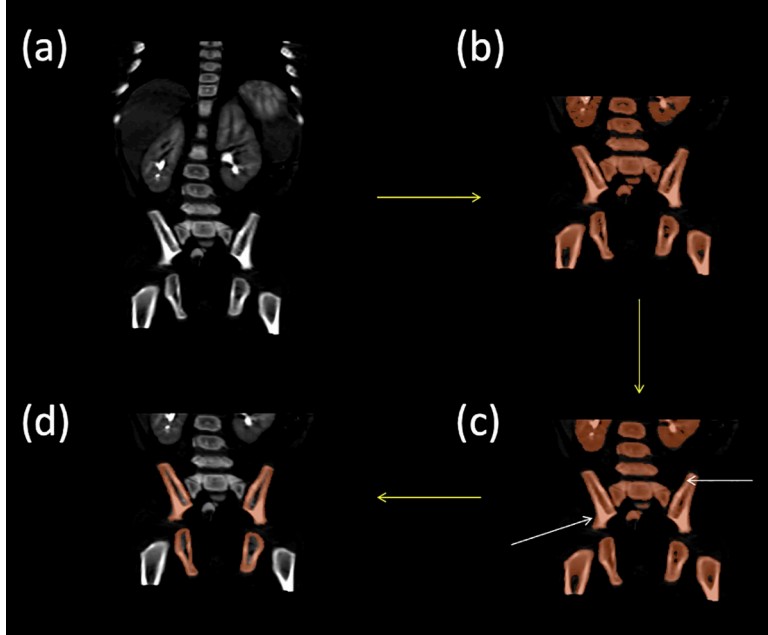

**Fig 5. Outlining the cortical bone segmentation process from CT.** (a) coronal section of the CT centered on the hips and pelvis, (b) the results of the semi-automatic threshold-based segmentation of the cortical bone (step ii), (c) the results of smoothing with a median filter with a kernel size of 5mm (step iii), and (d) the final segmentation result after manual refinement of the cortical bone (step iv).

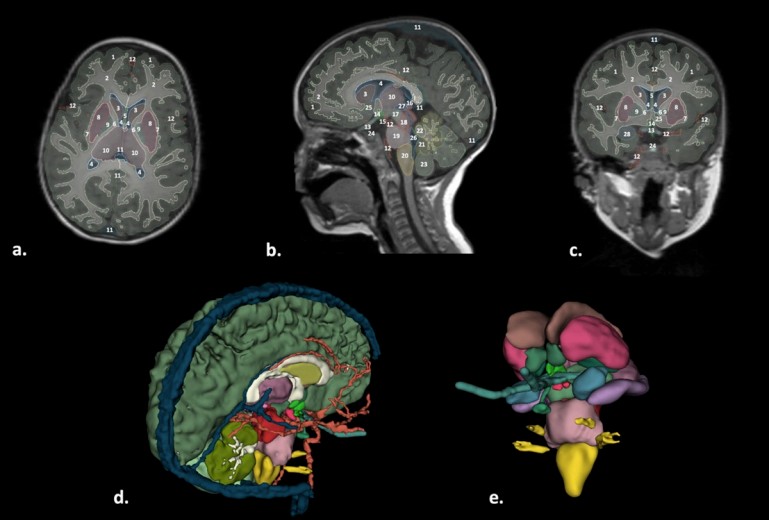

**Fig 6. Segmentation of brain structures (manual refinement of automatic segmentation) and vessels.** (a.) Axial, (b) sagittal, and (c) coronal views of the brain structures segmentations displayed on the MPRAGE. (d) and (e) show 3D surfaces of the brain and subcortical areas, respectively. Structures of interest: 1. Brain Grey Matter; 2. Brain White matter; 3, Caudate; 4. Lateral Ventricles; 5. Septum Pellucidum; 6. Internal capsule; 7. External Capsule; 8. Putamen; 9. Globus Pallidus; 10. Thalamus; 11. Cerebral Vein; 12. Cerebral Artery; 13b. Optic nerves; 13c. Optic nerves; 14. Hypothalamus and chiasm; 15. Mamillary bodies; 16. Epiphysis; 17. Ventral Diencephalon; 18. Midbrain; 19. Pons; 20. Medulla; 21. Vermis (Gray Matter); 22. Cerebellum White Matter; 23. Cerebellum Gray Matter; 24. Pituitary; 25. Nucleus Accumbens; 26. 4th Ventricle; 27. 3rd Ventricle; 28: Amygdala.

tissues that dominated the neighborhood using MATLAB (MathWorks, Natick, USA). The analysis and visualization of medical images were done with ANSA (BETA CAE Systems, Switzerland), 3D Slicer [46], and FreeSurfer [53–55].

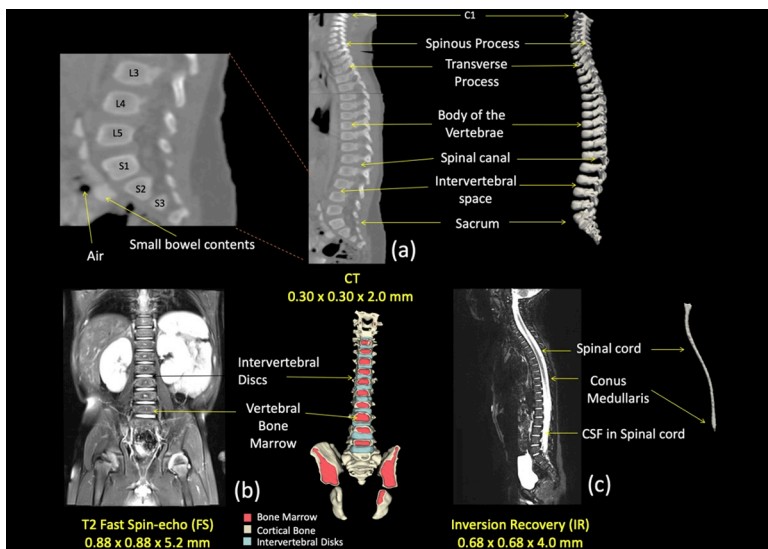

**Fig 7. Segmentation of the spinal cord and vertebrae.** (a) Sagittal view of the CT image used for the segmentation of the vertebrae. Arrows on the right were pointing anatomical structures included in the image and represented in the 3D reconstruction while anatomical locations were also marked on the left side. (b) Coronal view of the T2 image used to segment the intervertebral disks and the vertebral bone marrow, as shown with the 3D reconstruction on the right side. (c) Sagittal view of the IR used to delineate the spinal cord and the surrounding cerebrospinal fluid (CSF).

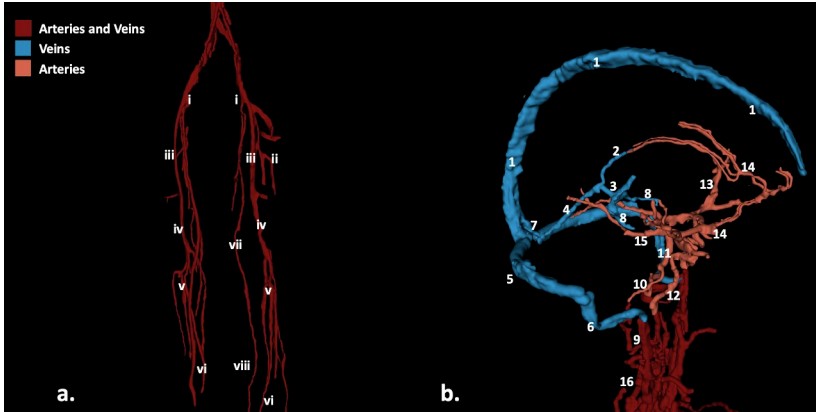

**Fig 8. Vessel segmentation in 3D representation.** a. Large arteries of the legs and their main branches. i. External Iliac Artery and vein, ii. Deep Femoral Artery and Vein of the thigh, iii. Femoral Artery and Vein, iv. Popliteal Artery, v. Peroneal vessels, vi. Anterior tibial vessels, vii. Great Saphenous Vein, viii. Lesser saphenous vein. b. Proximal large veins (blue) and arteries (red) of the brain and neck: 1. Superior Sagittal Sinus, 2. Internal Cerebral Vein, 3. Vein of Galen, 4. Straight Sinus, 5. Transverse Sinus, 6. Sigmoid Sinus, 7. Torcula, 8. Basal Vein of Rosenthal, 9. Internal Jugular Vein, 10.Vertebral Artery, 11. Basilar Artery, 12. Internal Carotid Artery, 13. Middle Cerebral Artery and main branches, 14. Anterior Cerebral Artery and main branches, 15. Posterior Cerebral Artery, 16. Arteries of the neck.

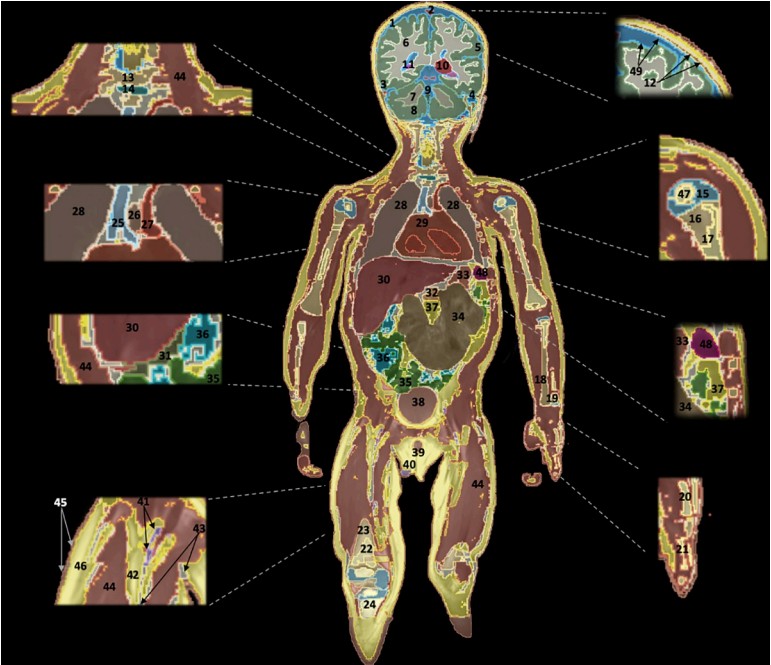

**Fig 9. Coronal view of the whole-body segmentation.** 1. Cerebrospinal Fluid, 2. Superior Sagittal Sinus, 3. Transverse sinus, 4. Air in the Mastoid, 5. Brain Grey Matter, 6. Brain White Matte, 7. Cerebellum White Matter, 8. Cerebellum Gray Matter, 9. Vermis, 10. Thalamus, 11. Hippocampus, 12. Skull, 13. Vertebrae, 14. Intervertebral Disc, 15. Humerus Cartilage, 16. Humerus Bone, 17. Humerus Bone Marrow, 18. Radius, 19. Ulna, 20. Metacarpal bone, 21. Proximal Phalanges, 22. Femur Bone Marrow, 23. Femur Bone, 24. Tibia, 25. Trachea, 26. Esophagus, 27. Aortic trunk, 28. Lungs, 29. Heart, 30. Liver, 31. Gallbladder, 32. Pancreas, 33. Stomach, 34. Large Intestines, 35. Small Intestines, 36. Air in the Small Intestines, 37. Intra-abdominal fat, 38. Bladder, 39.Penis (Corpus cavernosum, Corpus spongiosum), 40. Testis, 41.Vessels of lower extremities, 42.Fat, 43.Connective Tissues, 44.Muscle, 45.Skin, 46.Subcutaneous Fat, 47. Secondary Ossification Center, 48. Spleen, 49. Meninges.

## The validation

The Dice similarity coefficient (DSC) and Hausdorff distance were used to assess the inter-operator variability (**Fig 10**). The DSC allowed one to compute the level of agreement between different segmentations and was used to compare manual and automated image segmentation [56, 57]. Since DSC results can be biased in the evaluation of the segmentation of a large volume (i.e., liver) [58], we also estimated the Hausdorff distance, which measures the distance between two segmentations of the same tissue provided information on intra- and inter-operator variability. Both DSC and Hausdorff distance were measured in 11 different tissue compartments (Bladder, Right Humerus, Right Kidney, Spleen, Air, Vitreous Humour, Heart, Left Femur, Left Femur Cartilage, Liver, Lung) among the segmentations of the three trained segmentors (one more segmentor participated in providing statistical information). The resulting values were compared with labels already adjusted and confirmed by certified neuro-radiologists, P.E.G. and M.H.L. through an iterative process, which provided the most accurate

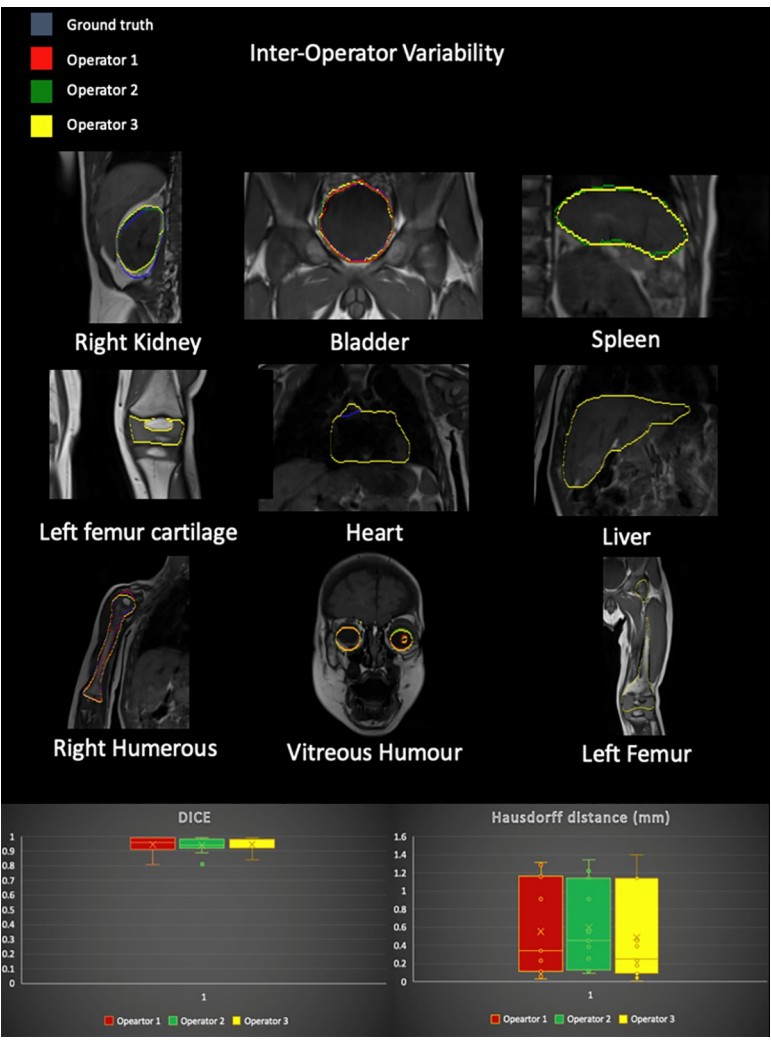

**Fig 10. Inter-operator variability.** (Top) Example of variability among segmentor A (red), B (green), and C (yellow) for a few representative structures in the body. (Bottom) Box plots of the values of DSC and Hausdorff distance for the 11 structures were included in the analysis. The variabilities were assessed by comparing each segmentation with a ground truth obtained via two CAQ subspecialized neuro-radiologists.

information of the MRI and CT segmentations without a bias. The following DSC definition was used:

$$DSC = \frac{2|X \cap Y|}{|X| + |Y|}$$

where X is the manually labeled region by one of our trained segmentors, Y is the region of interest volume that was confirmed as a ground truth by certified neuro-radiologists, P.E.G. and M.H.L., also through an interactive process [21]. DSC ranges from a value of 0 to 1, greater than 0.8 was considered as acceptable, whereas 0.9 was considered excellent.

The measured values of dimensions and volume of each tissue compartment were compared with literature values [28, 59–65]. Furthermore, in order to estimate the weight of each tissue compartment, we multiplied the tissue density taken from IT'IS database [23] with the volume of the tissue compartment. Segmented tissues were then adjusted and confirmed independently by certified neuro-radiologists, P.E.G. and M.H.L, also through an iterative process, following a "pass or fail" method. When both raters scored each tissue with "pass", the segmentation was finalized.

A Bloch simulation tool kit in Sim4Life [66] called SYSSIM was used in order to generate a synthetic MRI image for additional validation. The results of $B_1^+$ and E-fields at 3T (same as the base MR data) were resampled to the isotropic resolution of 0.8 mm × 0.8 m × 0.8 mm and fed into the T1- and T2- relaxation times and proton density tissue properties at 3T that were obtained from the literature [67–74]. The matching between the MRI image and the SYSSIM results was assessed using the mean-squared-error (MSE) and the cross-correlation.

### The pilot use case for MRI RF safety simulation

**Surface extraction.** Surface meshes were generated from a voxelized model by simplifying and smoothing of triangulated surfaces using a package Mesh toolkit in Sim4Life. In this process, we could achieve the reduction of model data size to an affordable level (reduction from 4.1 GB to 530 MB). Before the surface extraction, the segmentation project was finalized to the condition that any intersections between adjacent labels were removed by the Boolean subtract operator, and gaps between tissue labels were filled by the surrounding and prevalent tissue as described in the segmentation process section. Once a surface was extracted, curvature smoothing was applied using a constrained Laplacian surface smoothing method with small tolerance using geometric flow that prevents volume shrinking [75]. The methodology includes the self-intersection check function that any smoothing in the vicinity of self-intersection to be reverted and collapse the short edges and edge flips [16, 76]. Remaining self-intersection and non-manifold elements in the individual meshes were cured again using a Mesh Doctor tool in Sim4Life.

**Tissue properties.** Several studies have reported the variation of dielectric properties with age due to the change in the water content of tissues [77–79], resulting in both decreases in the value of the conductivity and permittivity with age. Previously, Peyman *et al.* reported changes in the conductivity across different rats ages due to a decrease in the tissue water content [79, 80]. The dielectric properties of 29-month-old tissues were estimated using the method of Dimbylow *et al.* [81] by scaling the adult tissue properties chosen from the IT'IS database [22, 23]. The conductivity and permittivity scaling factors for six reported tissues at 130 MHz in Peyman's study were computed as the ratio between adult rat and the ten days old rat, which corresponds to three human years for a rat [79, 80]. To estimate the scaling factor of non-reported soft tissue properties of the body averaged scaling ratio among four soft tissues was used except the skin and the bone that was treated as an outlier [36]. The dielectric properties

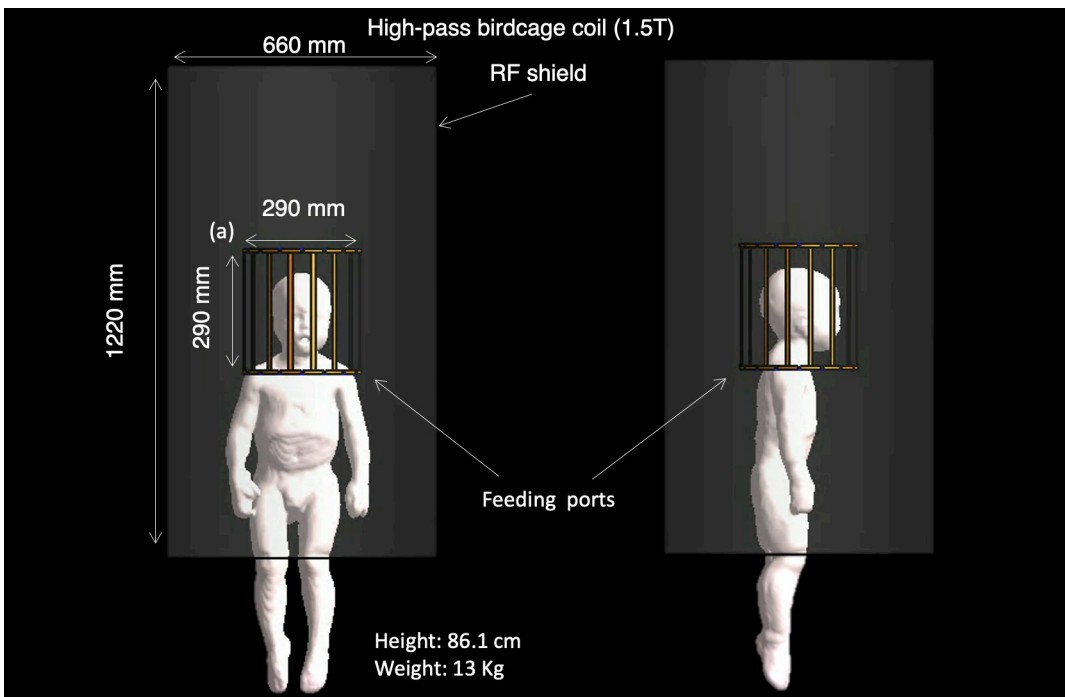

**Fig 11. The MARTIN pediatric model with head birdcage coil in 1.5 T.** The head of the model is positioned at the center of the high-pass birdcage coil tuned at 1.5 T.

of air, bile, blood, CSF, intestine contents, urine, aqueous humour, and vitreous humour were not accounted for changing with age [36, 81]. The conversion ratio between adult rats and ten days rat was estimated at 130 MHz, which was the lowest frequency measured in Peyman's study [79, 80]. According to J. Wells *et al.*, the variation of tissue density with age is relatively small for lean tissues (1.7% lower for the 5-year-old than adult lean tissue density) [82]. The tissue densities used in our simulations are from the tissue density database in IT'IS [23].

**A pilot use case for MRI RF safety simulation.** An MRI RF safety simulation was performed on MARTIN by implanting VNS electrodes to illustrate how safety simulation studies may be conducted. The simulation consisted of estimating the electric field (E-field) distribution and the specific absorption rate (SAR). Examples of simulations followed the newest (2018) International Organization for Standardization Technical Specification (ISO/TS) 10974 –assessment of the safety of magnetic resonance imaging for patients with an active implantable medical device [83]. Sim4Life (ZMT, Switzerland) was used to solve the Maxwell equation at 64 MHz using the finite-difference time-domain (FDTD) method [84]. 16-rung high-pass birdcage coil (diameter: 290 mm, length: 290 mm) tuned to 64 MHz ($S_{11}$ < -14 dB) and was used to generate a $B_1$ transmit field with circularly polarized mode with an RF shield [85] (**Fig 11**). MARTIN was used to calculate the EM field at 1.5 T, and the head was positioned at the center of the coil. The position and trace of the vagus nerve were registered from MIDA head and neck model [16] into MARTIN using the Elastix tool kit in 3D Slicer [46, 48]. Two cuff electrodes (cuff design with diameter: 1.4 mm, length: 1 mm, thickness: 0.3 mm) were positioned to surround the nerve, and the third anchoring VNS electrode was ignored in EM simulation (**Fig 12**). The lead trajectory followed the outline from the CT images and was also overseen by a neurologist, J.P. The insulation layer was added along with the lead, which entirely covered the lead. The diameter of the lead was 1 mm, and the insulation was 0.5 mm thick. The smallest size of the grid (0.3 mm) was set around the VNS implant to allow for a

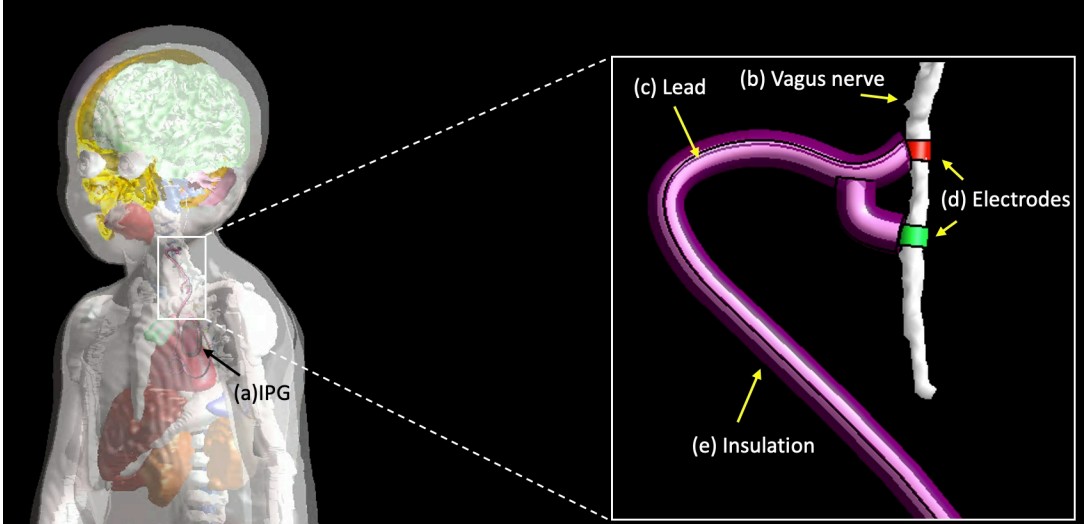

**Fig 12. The MARTIN pediatric model with the position of the VNS.** (a) pulse generator (IPG), (b) Vagus nerve, (c) lead, (d) Electrodes, (e) insulation (bottom).

continuous connection between electrodes and IPG while entirely wrapping the lead with insulation using a manual grid setting in Sim4Life. The dynamic grid resolution was used on the rest of the body, that had at least a 2 mm grid resolution on the body and a 1 mm grid resolution on the head [86]. The case without an implant was computed to estimate the contribution of the RF heating induced by the VNS implant. We studied the RF heating in a typical clinical scan [87] using the regional averaged SAR (in the head and whole-body), and 10g averaged SAR (10gSAR). The field magnitude was normalized to the $B_1^+$ ($B_1^+$ indicates the counter-clockwise rotational component of the transmit magnetic field) equals to 2μT at the coil center, to simulate a 90˚ flip angle with a rectangular pulse of 3 ms duration [88]. The EM fields were also normalized to the 3.2 W/kg SAR in the head to assess the RF heating, while we reported the rms E-field averaged over 10g mass [83].

## 3. Results

### The numerical model

The MRI data (**Fig 1**) from the T1 weighted, T2 weighted, MPRAGE, Inversion Recovery, HASTE, and VIBE allowed to successfully segment 86 tissues including brain structures, CSF, skull, vessels, eyes, kidneys, lungs, stomach, heart, muscle, skin, subcutaneous fat (SAT), spinal cord, cortical bone, bone marrows, and other soft tissues in the whole-body in MARTIN. **Table 3** shows the list of segmented tissue compartments, some of which are also shown in **Fig 9** with the coronal slice of the segmented tissues in the whole-body with the color-coded map.

 **Segmentation of the brain structures.** The results of automated-segmentation in the brain structures are shown in **Fig 4**, resulting in 17 different labels of the brain parenchyma (Brain Grey Matter, Brain White Matter, Cerebellum Grey Matter, Cerebellum White Matter, Ventricles, Thalamus, Caudate, Putamen, Pallidum, Hippocampus, Amygdala, Accumbens Area, Ventral DC, Vermis, Midbrain, Pons, Medulla) finely segmented (**Fig 4**; **S1 Table in S1 File**). High importance was given in achieving a highly detailed segmentation of the brain. As a result, the output of the automatic segmentation was manually adjusted, and some additional structures were included through an iterative process. Those structures were the cranial nerves

**Table 3. List of all tissue segmented.**

| | | | |
|---|---|---|---|
| Accumbens area | Eye Aqueous Humour | Meninges (Brain) | Skull |
| Adrenal gland | Eye Cornea | Meninges (Spinal cord) | Skull (Bone marrow red) |
| Air head | Eye Lens | Midbrain | Small Intestine |
| Air neck | Eye Muscle | Mucosa | Spinal cord |
| Amygdala | Eye Sclera | Muscle | Spleen |
| Bile | Eye Vitreous Humour | Muscle periocular | Sternum |
| Bladder wall | Fat between muscles | Nasal Cartilage | Stomach |
| Blood vessels body | Gallbladder | Optic Nerves | Sublingual glands |
| Blood Vessels (Circle of Willis) | Heart | Pallidum | Substantia Nigra |
| Bone Marrow (Red) | Hippocampus | Pancreas | Teeth |
| Bones (Cortical) | Hypothalamus | Penis | Testis |
| Brain Grey matter | Intervertebral discs | Pineal gland | Thalamus |
| Brain White Matter | Intestine Contents | Pituitary | Thymus |
| Caudate | Intestine Gas | Pons | Thyroid |
| Cerebellum Cortex | Intra-abdominal-chest FAT | Prostate | Tongue |
| Cerebellum White Matter | Kidney | Putamen | Trachea and main Bronchi |
| Connective tissue | Large Intestine | Rib and vertebrae (Bone marrow red) | Urine |
| Cranial Nerves (Large Branches II,V,VIII,IX) | Liver | Rib and vertebrae (Cortical bone) | Veins of Brain |
| CSF (Brain) | Long bones joint and femur cartilage | SAT (Subcutaneous fat) | Vermis Grey Matter |
| CSF (Spinal Cord) | Lung | Secondary ossification centers in long bones | Vermis White Matter |
| Epididymis | Mammillary body | Septum pellucidum | |
| Esophagus | Medulla | Skin | |

(CN II, V, VIII, IX), the hypothalamus, the pituitary gland, the pineal gland, the mammillary bodies, the white matter of the cerebellar vermis, the septum pellucidum, the veins and the arteries of the brain, the CSF and the meninges (**Fig 6**). The addition of the septum pellucidum to the segmentation affected the initial labels of the lateral ventricles, which were manually adjusted. Furthermore, the white matter was manually segmented for some gyri of the anterior frontal lobes and the external capsule, which also led to the manual refinement of the neighboring part of the putamen. The adjustment of the posterior genu of the internal capsule led to some changes in the automatic segmentation result in the globus pallidus and the thalamus bilaterally. White matter tracts from the cerebellar vermis and the lateral hemispheres were also manually added using the 3D Slicer software (**Fig 6**).

**Segmentation of the non-brain structures.** Semi-automatic techniques followed by manual adjustment through an iterative process were applied for the segmentation of all the tissues of MARTIN, except for the brain structures, the segmentation process of which was described above (**S2 and S3 Tables in S1 File**). Multiple sequences were used for the more challenging type of tissues as the blood vessels running inside the skull for which T2 Flair images were first consulted. The result was superimposed onto the T1 weighted MPRAGE sequence that was used for the segmentation of the brain structures. The final adjusted result was projected onto the whole body T1 in order to ensure the alignment with the rest of the tissues of the head (**Figs 6 and 8**). A knowledge-based segmentation was done for parts of the vessels that were lacking given MRI contrast. VIBE sequence with contrast was used to keep track of the vessels of the lower extremities during the segmentation, of which we did not differentiate the arteries from the veins. The tissues were finalized after achieving intra-operator agreement, inter-operator agreement, and once they were adjusted and confirmed by the two neuro-radiologists through an interactive process and were well aligned with the neighboring tissues (**Fig 2**). **Fig 8** shows a 3D reconstruction of the volume of the vessels of the brain, the neck, the thorax, the abdomen, the pelvis, and the lower extremities

## The validation

Brain CSF was initially segmented based on the IR sequence of the head. The initial CSF segmentation was combined and reevaluated with the segmentation of the neural brain structures and the vessels of the brain using the MPRAGE sequence. The final result was adjusted on the T1 Spin Echo sequence to ensure the alignment with the bony structures of the head. After segmenting the meninges, outlining the border of the CSF with the brain and skull, the volume of the CSF was calculated to be 108.7 cm$^3$, 1.2% smaller than the literature values [60] (**Table 4**).

The heart was segmented on the whole-body T1 and was computed to be 56.7 g, which is 0.01% larger than the literature value. The liver was also segmented on the whole body T1 and after outlining its border with the gallbladder (wall and bile) as well as after segmenting the big vessels running through it. Its volume was estimated to be 335 cm$^3$, which is within the literature range of values (0%) for the age of MARTIN [61]. Regarding the spleen, the abdominal IR was first consulted, and the final segmentation was performed on the whole body T1, having a volume of 38.5 g, which is 4.1% larger than the literature value, while the longitudinal diameter had no difference (0%) from the literature values [62]. The same process was followed for the kidneys for which the left and the right kidney weighed 48 g and 48.6 g, respectively, being 4.2% and 3.3% larger than the literature values. Adrenal glands were segmented after the segmentation of the kidneys, and their combined weight was 6.19 g, which is only 0.01% different for the reference values [89]. IR of the chest was first used for the segmentation of the thymus, and the tissue was finalized using the whole-body T1, with its weight being 19.8 g, which is only 0.1% smaller than the reference value. The same technique was used for the segmentation of the testis for which the combined weight was 2.9 g that had no difference (0%) from the reference value [63]. The sternal bone, which was segmented using the CT had no (0%) difference from the reference values.

Since a CT was not available for the extremities, the long bones were segmented using MRI data. Some of the model's measurements that were used for the adjustment and validation of the segmentation process are shown in **Table 4**. In particular, the bone length ratio for the humerus to the radius had 3.9% difference from the literature values, for the tibia to the femur,

**Table 4. Validation table (absolute value of the difference between the measured value and literature value).**

| Tissue | Measurement type | Measured value | Literature value | Difference (%) |
|---|---|---|---|---|
| Adrenal glands (combined) | Weight (g) according to age | 6.46 g | 6.2–6.7 g [28,59] | 0% |
| Brain | Weight (g) according to age | 1052 g | 1064 g [28] | 1.1% |
| Brain CSF | Volume (cm$^3$) according to age | 108.7 cm$^3$ | 110–120 cm$^3$ [60] | 1.2% |
| Heart | Weight (g) according to age | 56.7g | 56 g [28] | 0.01% |
| Liver | Volume (cm$^3$) according to age | 335 cm$^3$ | 299.7–426.2 cm$^3$ [61] | 0% |
| | Longitudinal dimension (mm) according to age and height | 82.3 mm | 85 mm [62] | 1.6% |
| Spleen | Weight (g) according to age | 38.5 g | 37 g [28] | 4.1% |
| | Longitudinal dimension (mm) according to age and height | 70 mm | 70 mm [62] | 0% |
| Right Kidney | Weight (g) according to age | 48.6g | 47g [28] | 3.3% |
| | Longitudinal dimension (mm) according to age and height | 65.8 mm | 61 mm [62] | 3.8% |
| Left Kidney | Weight (g) according to age | 48 g | 46 g [28] | 4.2% |
| | Longitudinal dimension (mm) according to age and height | 68.8 mm | 66 mm [62] | 2% |
| Thymus | Weight (g) according to age | 19.8 g | 20–38 g [28] | 0.1% |
| Testes (combined) | Weight (g) according to age | 2.9 g | 2.3–4.3 g [63] | 0% |
| Sternal Bone | Length (cm) according to age | 10.2 cm | 8–12.5 cm [64] | 0% |
| Radius/Humerus | Bone length ratio according to age | 0.78 | 0.75 [65] | 3.9% |
| Tibia/Femur | Bone length ratio according to age | 0.81 | 0.81 [65] | 0% |
| Humerus/Femur | Bone length ratio according to age | 0.71 | 0.71 [65] | 0% |
| Radius/Tibia | Bone length ratio according to age | 0.69 | 0.65 [65] | 5.9% |

**Table 5. Inter-operator variability across structures on the coronal slices.**

| Tissue type | Dice Similarity Coefficient (DSC) | | | Hausdorff distance—average (mm) | | |
|---|---|---|---|---|---|---|
| | 1 vs. GT | 2 vs. GT | 3 vs. GT | 1 vs. GT | 2 vs. GT | 3 vs. GT |
| Bladder | 0.91 | 0.92 | 0.92 | 1.31 | 1.14 | 1.14 |
| Right Humerus | 0.81 | 0.81 | 0.84 | 1.16 | 1.22 | 1.17 |
| Right Kidney | 0.89 | 0.89 | 0.89 | 1.29 | 1.34 | 1.40 |
| Spleen | 0.99 | 0.93 | 0.93 | 0.03 | 0.38 | 0.39 |
| Air head and neck | 0.96 | 0.94 | 0.98 | 0.23 | 0.55 | 0.18 |
| Vitreous Humour | 0.99 | 0.98 | 0.98 | 0.06 | 0.12 | 0.09 |
| Heart | 0.93 | 0.93 | 0.98 | 0.91 | 0.91 | 0.23 |
| Left femur | 0.98 | 0.97 | 0.99 | 0.34 | 0.45 | 0.04 |
| Left femur cartilage | 0.94 | 0.98 | 0.92 | 0.36 | 0.13 | 0.45 |
| Liver | 0.98 | 0.98 | 0.98 | 0.26 | 0.25 | 0.25 |
| Lung | 0.99 | 0.99 | 0.99 | 0.11 | 0.09 | 0.01 |

and for the femur to the humerus, the ratio had no difference (0%) from the literature and from the radius to the tibia the ratio was calculated 5.9% for the age of the patient.

The way that DSC was integrated into our validation system is demonstrated in the flow-chart in **Fig 2**. The inter-operator variability indices (**Fig 7**) were calculated for each segmentor independently and compared against the ground truth that was adjusted and confirmed by certified neuro-radiologists through an iterative process [21]. The example of the segmentations done by segmentor A (red), B (green), and C (yellow) for nine representative organs in the body is shown in **Fig 10**. The results of DSC and Hausdorff distance in 11 tissue compartments were ranked as DSC = 0.94 ± 0.05 (mean ± SD) and HD = 0.54 ± 0.48 across segmentors and over the representative tissue compartments (**Table 5**). The highest match (DSC > 0.99 and HD < 0.11) for all the comparison was found in the lung, and the lowest (DSC > 0.81 and HD < 1.22) was found in the right humerus. The evaluation of the DSC showed that the intra-operator variability was less than 20%, which indicated that the intra-operator bias was marginal (**Table 5**). The results of a Bloch Simulator (SYSSIM) were compared with the original inversion recovery MRI data in **Fig 13**. The MSE between the original IR sequence and SYSSIM returned the value of 0.14 (in normalized and zero averaged intensities), which along with the cross-correlation coefficient of 0.782 (See S3 Fig in S1 File for the cross-correlation plots) indicates a high level of matching between the two images.

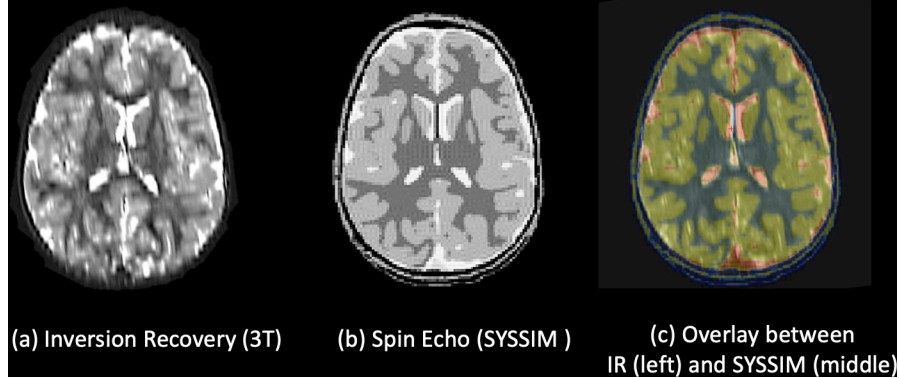

**Fig 13. SYSSIM results.** (a) shows the axial view of the Inversion Recovery used in segmentation, (b) shows the results of Bloch simulation, SYSSIM, using proposed voxel model, (c) shows the overlay between MRI image ((a), in grayscale) and SYSSIM ((b), in colormap scale).

## Pilot use case for MRI RF safety simulation

The dielectric properties of MARTIN's tissues at 64 MHz and 128 MHz were shown in **Table 6**. The highest permittivity ratios (i.e., adult rat vs. ten days old rat) were found in the bone, the brain, and in the skin as 1.99, 1.33, and 1.29, respectively, compared to the averaged soft tissue ratio of 1.22. The conductivity ratios of 2.4, 1.6, and 1.5 were calculated for the bone, the brain, and the skin, respectively. The tissue property assignment for the tissues for which no measurements have been published was shown in **S4 Table in S1 File** [16, 21, 23]. We studied the tissue density variation of 1.7% in the uncertainties table in (S5 Table in S1 File), and found that the uncertainty budget is very small, $< 0.03\%/\%$, thus can be neglected [36]. The head averaged SAR ($SAR_{head}$), whole-body averaged SAR ($SAR_{wb}$), 10gSAR, and 10g averaged rms E-field (10gE) of a pilot use case of MARTIN with and without a VNS implant are reported in **Table 7**. The marginal difference was found in $SAR_{head}$ between MARTIN in 1.5T without an implant (0.2046 W/kg) and in the case with a VNS implant (0.2064 W/kg) when fields were averaged to 2μT at the center of the coil [88]. Similar results were found at the $SAR_{wb}$, which were 0.0456 W/kg for MARTIN without an implant, 0.0462 W/kg for in the case with a VNS implant. The results of a maximum 10gSAR were analyzed to assess the RF heating in normal MRI operational mode [87] that reported 47% higher results of 10gSAR (0.7137 W/kg) when a VNS implant was included in the simulation compare to the case without an implant (0.4858 W/kg). **Fig 14** shows the maximum intensity projection of 10gSAR with and without a VNS implant. The highest 10gSAR was found in the side of the neck without a VNS implant and near the VNS electrodes in the case with the VNS implant. In the case of RF exposure limit in the normal mode head MRI, the maximum 10g|$E$| in the head was shown 281.05 V/m near the electrodes in the case of VNS, which in this pilot example case. (See the uncertainties analysis shown in **S5 Table in S1 File**, and an example of incident tangential E-field calculation using for Tier 3 analysis in **S1 Fig in S1 File**). The uncertainties analysis based on Martin with a VNS implant shows a maximum 10g E-field sensitivity with Skin conductivity (See S5 Table in S1 File). Finally, we compared the results of head averaged SAR from Martin with the results using Nina model (Table 7), which were 0.2046 W/kg and 0.2068 W/kg, corresponding only to marginal variations of 1.06%. (See S7 Table in S1 File for the simulation results comparison between Martin and Nina with adult tissue properties [23]). Even though the variations are marginal, the anatomical inaccuracies of Nina not present in MARTIN, may lead to SAR peaks in mis-labelled tissue.

## 4. Discussion and conclusion

MARTIN model is developed as a computational model that has full body anatomical detail, and that can be primarily used for investigating tissue interactions with electromagnetic fields generated from the medical device as well as for the dosimetry studies of children around 29-month-old. An advantage of this study is the use of in-vivo medical images of a 29-month-old child. Such images represent the age-appropriate development of the different tissue, such as the brain, the CSF, the heart, the liver, the skull, the bone marrows, the finger bones, and more that were difficult to account correctly when the model is morphed from older due to the various development rate in each organ (**Table 4; S2 Fig in S1 File**). The validation methods of the model include an adjustment and confirmation by two neuroradiologists through an interactive process to avoid personal bias [16, 21], as well as the metrics of physical measurement (e.g., volume, length, weight) [31, 91]. The segmentation of the skull and deep-brain structures may be useful for transcranial focused ultrasound modeling [92] and neuromodulation application [93], and the presence of blood vessels can be valuable for accurate thermal estimation in RF safety studies in MRI [94, 95].

**Table 6. Dielectric properties of 29-month-old at 1.5T and 3T.**

| Tissue | Permittivity ratio [a] | 29-month-old tissue Permittivity | | Conductivity ratio [a] | 29-month-old tissue Conductivity (S/m) | |
|---|---|---|---|---|---|---|
| | | 64 MHz | 128 MHz | | 64 MHz | 128MHz |
| Adrenal Gland | 1.15 | 74.73 | 73.36 | 1.4 | 0.8810 | 0.8956 |
| Air | 1 | 1 | 1 | 1 | 0 | 0 |
| Bile | 1.0 | 105.44 | 88.9 | 1.0 | 1.4818 | 1.5764 |
| Blood | 1.0 | 86.44 | 73.16 | 1.0 | 1.2067 | 1.2486 |
| Blood Vessel Wall | 1.22 | 83.74 | 68.31 | 1.4 | 0.6010 | 0.6705 |
| Bone (Cortical) | 1.99 | 33.2 | 29.29 | 2.4 | 0.1429 | 0.1616 |
| Bone Marrow (Red) | 1.22 | 20.05 | 16.52 | 1.4 | 0.2161 | 0.2268 |
| Brain (Grey Matter) | 1.33 | 129.58 | 97.78 | 1.6 | 0.8174 | 0.9388 |
| Brain (White Matter) | 1.33 | 90.22 | 69.87 | 1.6 | 0.4664 | 0.5474 |
| Cartilage | 1.22 | 76.76 | 64.57 | 1.4 | 0.6329 | 0.6837 |
| Cerebellum | 1.33 | 154.75 | 106.05 | 1.6 | 1.1504 | 1.3270 |
| Cerebrospinal Fluid | 1.0 | 97.31 | 84.04 | 1.0 | 2.0660 | 2.1430 |
| Connective Tissue | 1.22 | 72.58 | 63.27 | 1.4 | 0.6641 | 0.6982 |
| Dura | 1.33 | 97.44 | 74.44 | 1.6 | 1.1307 | 1.2027 |
| Eye (Cornea) | 1.22 | 106.6 | 87.18 | 1.4 | 1.4008 | 1.4822 |
| Eye (Lens) | 1.22 | 61.41 | 52.21 | 1.4 | 0.4002 | 0.4378 |
| Eye (Retina) | 1.33 | 129.58 | 97.78 | 1.6 | 0.8174 | 0.9388 |
| Eye (Sclera) | 1.22 | 91.87 | 79.3 | 1.4 | 1.2357 | 1.2847 |
| Eye (Vitreous Humor) | 1.22 | 84.33 | 84.26 | 1.4 | 2.1044 | 2.1075 |
| Fat | 1.22 | 16.65 | 15.09 | 1.4 | 0.0926 | 0.0976 |
| Gallbladder | 1.0 | 87.4 | 74.14 | 1.0 | 0.9660 | 1.0418 |
| Heart Muscle | 1.18 | 125.69 | 99.42 | 1.4 | 0.9498 | 1.0726 |
| Intervertebral Disc | 1.22 | 62.78 | 60.63 | 1.4 | 1.1807 | 1.2043 |
| Kidney | 1.22 | 144.64 | 109.33 | 1.4 | 1.0378 | 1.1932 |
| Large Intestine | 1.22 | 115.49 | 93.42 | 1.4 | 0.8934 | 0.9873 |
| Liver | 1.22 | 98.28 | 78.39 | 1.2 | 0.5376 | 0.6131 |
| Lung | 1.22 | 45.26 | 35.95 | 1.4 | 0.4046 | 0.4419 |
| Muscle | 1.18 | 85.24 | 74.92 | 1.4 | 0.9635 | 1.0069 |
| Nerve | 1.22 | 67.18 | 53.76 | 1.4 | 0.4370 | 0.4953 |
| Salivary Gland | 1.15 | 92.71 | 91.38 | 1.4 | 0.9507 | 0.9644 |
| Skin | 1.29 | 118.9 | 84.41 | 1.5 | 0.6536 | 0.7841 |
| Small Intestine | 1.22 | 144.4 | 107.33 | 1.4 | 2.2280 | 2.3700 |
| Spleen | 1.22 | 134.88 | 101.13 | 1.4 | 1.0415 | 1.1693 |
| Stomach | 1.22 | 104.7 | 91.37 | 1.4 | 1.2290 | 1.2779 |
| Tendon\Ligament | 1.22 | 72.58 | 63.27 | 1.4 | 0.6641 | 0.6982 |
| Testis | 1.22 | 103.12 | 88 | 1.4 | 1.2388 | 1.2970 |
| Thymus | 1.22 | 68.09 | 66.95 | 1.4 | 0.8924 | 0.9032 |
| Thyroid Gland | 1.15 | 85.04 | 76.8 | 1.4 | 1.0896 | 1.1258 |
| Tongue | 1.22 | 91.87 | 79.3 | 1.4 | 0.9130 | 0.9620 |
| Trachea | 1.22 | 71.85 | 61.7 | 1.4 | 0.7398 | 0.7831 |
| Ureter\Urethra | 1.22 | 83.74 | 68.31 | 1.4 | 0.6010 | 0.6705 |
| Urinary Bladder Wall | 1.22 | 30.01 | 26.67 | 1.4 | 0.4023 | 0.4172 |
| Urine | 1.0 | 49.95 | 49.95 | 1.0 | 1.7500 | 1.7500 |

a. The permittivity (conductivity) ratio is the permittivity of the 29-month-old tissue/permittivity of adult tissue [23, 36, 79]. Dielectric parameters were based on Gabriel dispersion relationship [90] and IT'IS database [23]. Values from similar tissues were assigned for tissues for which no measurement value have been published [16, 23] (also see **S4 Table in S1 File**).

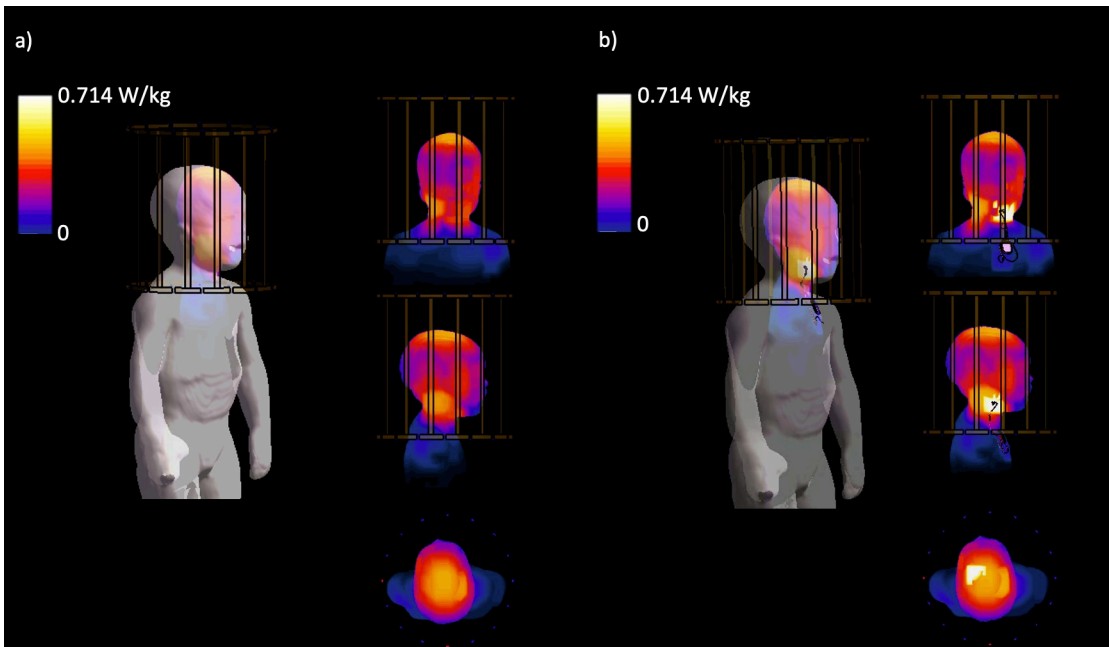

**Fig 14. Results of 10g averaged SAR.** a) Maximum intensity projection of 10gSAR without an implant in 3D view and the coronal, sagittal, and axial view, b) Maximum intensity projection of 10g averaged SAR in the case of a VNS implant in 3D view, and the coronal, sagittal, and axial view (*EM simulation results were averaged to 2µT at the coil center [82]).

The discussion follows the order of the workflow for the development, validation, and the pilot use case of the model.

## The numerical model

**Image acquisition and pre-processing.** A well-designed image acquisition plan would allow the optimal condition of data required for the tissue segmentation [16]. However, the special considerations on the pediatric population discourage unnecessary imaging due to the potential need for sedation for motion control [96], and the risk on the pediatric subject against radiation exposure [97]. In regard to MRI, the high-resolution and multiple image

**Table 7. Specific absorption rate and 10g averaged rms E-field results of Nina and MARTIN at 1.5T using Tx/Rx head coil.**

| | | Nina in 1.5 T using Head Tx/Rx coil (Head centered, No implant) | MARTIN in 1.5T using Head Tx/Rx coil (Head centered, No implant) | MARTIN in 1.5T using Head Tx/Rx coil (Head centered, with a VNS implant) |
|---|---|---|---|---|
| Fields were averaged to 2µT at the center of the coil [87] | Head averaged SAR (W/kg) | 0.21 | 0.20 | 0.21 |
| | Whole-body averaged SAR (W/kg) | 0.04 | 0.05 | 0.05 |
| | Whole-body maximum 10gSAR (W/kg) | 0.48 | 0.49 | 0.71 |
| | Normalization factor (V) | 13.71 | 13.95 | 14.04 |
| Fields were averaged to 3.2 W/kg in the head (Scanner head scanning limit) [83] | Maximum 10g rms E-field (V/m) | 176.54 | 165.45 | 281.05 |
| | Normalization factor (V) | 53.94 | 55.17 | 55.26 |

scans used in the production of high-resolution anthropomorphic adult models such as MIDA [16] might not be an option in pediatric models. In particular, biological noise, anatomical posture [98] (due to smaller bodies), motion [99], and behavioral performance (especially in early age) may significantly affect the quality of pediatric models [99, 100]. Hence existing scans in the data repositories such as the PACS medical technologies available in Hospitals were used instead, which limits the available images (e.g., clinical images often lack isotropic resolution). For this reason, most of the MRI sequences used in this study were not acquired with the iso-resolution in all three-directions. The resolution of segmentation is binding to the base medical image; thus, data resampling was required for all medical images used in segmentation. Isotropic resolution of 0.5 mm was chosen; in this way, anatomical information captured in submillimeter resolution was accounted for details of small geometric changes while the stair-cased effect between slides can be minimized.

**Data co-registration.** Data co-registration was required due to the use of images with multi-regional sequences and different modalities. Inter-modal data registration was relatively straightforward in the data acquired from the same subject. For example, MR images were acquired in a single session where a child subject remained in the same posture. Thus, most of the registration was related to the simple translation and rotation in 3D (x-, y-, z-). Whereas, the registration of CT scan into MRI required a non-linear registration [48] on three CT subsections: neck from C1 to T5 vertebrae, the chest between T6 and L5 vertebrae, and abdominal below L5 vertebrae. The partial-body CT scan was conducted 5-month before the MR scan that caused size difference within intra-subject data, and also the position of the subject, which made it difficult to register the entire CT images into MRI at once. CT was still valuable information to capture the boundary of calcified bone within the coverage of scan (e.g., hands and legs were missing) followed by the final manual refinement based on the T1 whole-body completed the registration. The age of the model was determined by the time of the MRI scan, as it was used for most of the segmentation and final refinement.

**Data segmentation and labeling.** The critical process of model development is data segmentation and labeling. This includes the classification of the boundary of the tissue and the assignment of the correct single label to each region based on anatomical atlases. Automatic, semi-automatic, and manual segmentation can be used, and the accuracy of the segmentation was determined by a function of the base image resolution, the contrast offered by sequence and modality, the signal-to-noise ratio, the presence of artifact (motion and system), and the co-registration of data. The automated segmentation process offers a reduced time of work and also provides an opportunity for one to separate anatomical regions with limited knowledge of anatomical atlases (e.g., separation of white matter and gray matter using T1 and T2 contrast in MRI). Although, the accurate segmentation of complex structures was challenging to be solely determined by an algorithm without continued supervision and feedback by an expert in anatomical atlases due to unclear boundaries, signal inhomogeneities, and noises. Thus, an addition of knowledge-based refinement on automatic segmentation was required.

## The validation

The adjustment and confirmation using DSC and the Hausdorff distance estimation during segmentation and the oversight by neuro-radiologists were extremely valuable as feedback in the editing loop that led to minimizing the errors in labeling and the accurate segmentation of the boundaries of various tissues. Furthermore, the adjustment and confirmation with metrics from literature (**Table 4**) show the great agreement in terms of length, volume, and mass. Although these results were not made by a single trial of segmentation, they were rather achieved after the multiple stages of trials and errors and interactive processes (**Fig 2**). For

instance, the segmentation of CSF in the brain was not trivial due to the close proximity of the extra-axial CSF to the brain tissue and the skull. The boundary of CSF was initially estimated by IR data, which led to an overestimation of the CSF region of boundary since the contrast changes between CSF and grey matter were not as dramatic as the T1-weighted sequence. As a result, CSF took over the cortex of the brain when two labels were inspected together in the T1-weighted reference image. Alternatively, we followed the previous study [101] that suggested a method to fill all the empty space between the skull and the cortex of the brain, which only fitted to the value when CSF and meninges considered as a single label [101] but deviated from other literature values that CSF and meninges were considered separately (12.2% difference) due to the presence of meninges on the CSF label. To tackle this issue, we created a thin layer of tissue outlining the inner surface of the skull and the surface of the brain cortex, and we subtracted this tissue layer from the cumulative label that was generated in the filling method. After separating the meninges from the CSF, the discrepancy between the measured volume of CSF and the literature value was only 1.2%. As a final verification step of the CSF segmentation, the initial segmentation done via IR data was used to confirm the anatomical location of our CSF label in the 3D space. The improvement in CSF segmentation results highlights the importance of adjustment and confirmation against the values reported in the literature. Both the brain and the CSF of the head volumes were calculated to have slightly lower values compared to the literature (1.1%, and 1.2%, respectively), which supports the accuracy of their relative size. It is also important to mention that the reference used provided absolute values for the weights of those tissues and not a normal range, which implies that it is highly likely that the scans belonged to a subject with a slightly smaller head size.

The CT data were a useful source to determine the boundaries of cortical bones. However, skull boundary detection was challenging since the CT images did not fully cover the head due to the safety limit made to protect a child's brain exposure against ionizing radiation. Thus, the upper skull boundaries were extended from the CT of the mandible and required the aid of dedicated skull segmentation tools as a starting point of segmentation [52, 102]. An extensive amount of manual refinement on the skull was still required due to marginal errors on registration between MR and CT, and the estimation done by the automated skull segmentation tool. The process of filling the unexpected holes in the skull along with the inspection of the relationship of the skull with the neighboring tissue labels (e.g., SAT, CSF) was done thoroughly several times. The results of the bone segmentation could have been improved if only the CT data included the extremities by design. Similar to the skull segmentation, knowledge-based segmentation using the whole-body T1 MRI sequence was done for the segmentation of the small-sized bones, such as the carpal, and tarsal bones, fingers, and toes where CT scans did not cover these areas. The small-sized bones lack clear contrast in MR sequences (e.g., T1-weighted, IR sequences), which showed a degree of limitation as to the relative lengths of the radius/tibia (5.9%), and radius/humerus (3.9%) segmentation shown in the confirmation process using literature values in **Table 4**. Multiple references were used for hand bones and other regions and other limbs [65, 103, 104] to increase the confidence in the segmentation of extremity bones in MR data (**S2 Fig in S1 File**). Given the limitations mentioned above, we achieved a high-quality segmentation, as shown by the relative sizes of the humerus/femur and the tibia to the femur, which showed no difference (0%) compared to literature values. The detail of the process followed was also represented by the result of the segmentation of the bones of the hand, which matched the age of the child when compared to the gold standard of the digital atlas of skeletal maturity [104]. Also, unique pediatric tissues specific for this age were segmented as the ossification centers of the long bones.

## Limitation on segmentation

Limitations of our study include the resolution along the z-direction of the model (toe to head) and the segmentation of boundaries between organs. The base image resolution along the z-direction was the range between 1.0 mm to 5.2 mm. Acquiring high-resolution data requires a proportionally longer amount of scan time; for that reason, the state-of-the-art fast scanning method may be considered in order to acquire the image for our segmentation project [105–108]. The segmentation of boundaries between organs was challenging despite the available specialized sequences. All the sequences were co-registered to the whole-body T1 image. However, it was difficult to achieve 100% accuracy of registration due to the physiological noise, artifacts from motion, different spatial and contrast resolution between sequences, and the lack of signal in certain regions such as a nasal cavity, CSF, cortical bones, and skin. As a result, a requisite amount of time was invested into the adjustment of segmentation and inspecting any overlap of tissue labels that were initially segmented using different sequences. Vessel segmentation is an excellent example of a challenge on the intersection between segmentation and boundary detection as it is touching and travels through the various organs in the body. Also, the CT data has a 5-month time lag compared to the MR data as the MR and CT images used for the segmentation were clinical recordings obtained from the PACS system of Boston Children's Hospital. We could not find in our PACS system a patient without a distinctive disease that could affect its anatomy to have whole-body MRI and also CT scans at the same time. The CT data were morphed to aid manual segmentation of the mandible, the vertebrae, and the rib cage, while the MRI scans were used to segment most of the bones, such as the Radius, Tibia, Ulna, and many others (see S2 Table in S1 File). All of this initial segmentation was reevaluated and modified on based on the reference MR data as the CT data were not perfectly aligned in a few spots due to the different time of the scan, the posture of the subject, and the respiration state (e.g., misalignment of the ribcage around the lungs). As a result, a considerable amount of time was required to inspect visually, and further adjustment was required between different segmentation labels on the T1-reference image. After segmenting the fat located subcutaneously, intra-abdominally, and between the muscles, we assigned the label of connective tissue to the empty spaces located mainly in the abdomen and the thorax. MATLAB was used to inspect any remaining unlabeled region outside the background and to assign the label that was surrounding the empty voxel with the highest percentages. The two neuroradiologists confirmed both the process of the segmentation in the initial sequence as well as the process of the adjustment of each tissue fused together in the reference T1 and the post-processed label after filling the unlabeled space with an interactive process. Another limitation is based on the Nyquist phenomenon. Specifically, we employed a knowledge-based segmentation for tissues difficult-to-identify on the MRI, such as the adrenal glands (since resampling the spatial image resolution to 0.5 mm × 0.5 mm × 0.5 mm is insufficient for detection and delineation). The Nyquist phenomenon, therefore, reflects limitations imposed by the original spatial resolution of the scanned images. Finally, segmentation of small tissue structures, such as the upper extremities vessels, was challenging due to unavoidable scan data limitations. Despite this need for knowledge-based segmentation, however, our final adjusted and confirmed results all fell within appropriate published literature values (**Table 4**).

## Pilot use case for MRI RF safety simulation

**Surface mesh extraction.**    Several open-source tools were initially attempted to generate meshes, such as iso2mesh [109], and Quality Multi-Domain Meshing with No Gap (QMDMNG) tool developed by UT Austin [110], which were not suitable to cope with the whole-body high-resolution anatomical model. A specially developed algorithm in Sim4Life is

a tool to extract surface meshes, which are topologically conformal and high-quality triangulated elements [16]. The mesh processing routine in Sim4Life was introduced during the development of "Virtual Population 3.0" [21] that we were able to take full advantage of a dedicated mesh generation tool. Before processing the mesh, the extensive amount of manual refinements was needed to clean up noise, segmentation artifacts, and dealing with unwanted holes at the segmentation stage, which was the essential requirement to produce smoothed unstructured tetrahedral mesh surface with anatomical fidelity [16]. More details of the discussion about alternative surface mesh extraction techniques can be found in Makarov *et al.* [17].

**Pilot use case for MRI RF safety simulation.** $SAR_{head}$, $SAR_{wb}$, and 10gSAR in 1.5T reported in this study with MARTIN and Nina without an implant were consistent with that of the Baby model developed by Helmholtz Zentrum München [29] reported in 2015 by Malik *et al.* [111], which was a valuable reference to compare the simulation setup and use of dielectric properties conversion. The MRI labeling guidelines of commercial VNS (Cyberonics, Houston, TX) [42] suggest their VNS product only to be used in the condition of localized RF filed exposure above or outside of the C7 and T8 vertebrae where the VNS leads, and the implantable pulse generator (IPG) is usually located. However, this level was only investigated in local Tx/Rx coil, which showed that with specific pathways of the leads and different coil geometry, there is a chance to increase the heating [112]. Such various scenarios of simulation require a large amount of calculation time due to a dense simulation grid on an elongated active implant device. For this reason, device response from a lab-measurement can be incorporated into the simulated incident tangential E-field results (without the device) to estimate the power deposition in multiple potential lead trajectory scenarios, which is called Tier-3 analysis proposed on ISO/TS 10974 (The example of the use of our MARTIN model in Tier-3 analysis is shown in **S1 Fig in S1 File**). The scope of our simulation did not account for inter-subject variabilities such as body size, shape, gender, and ages. According to the FDA recognized standard, MRI guidelines for active implants TS 10974:2018, further simulations and measurements are needed such as the different trajectory of implant lead, using various voxel models, and studying different dimension of the body RF transmit coil [83], which is particularly important in children who have more space to move in an MRI. The 10gSAR safety margin of 1.5 was suggested by Garrec *et al.* [113] to account for the inter-subject variability in MRI RF safety assessment to guarantee that there is a chance of less than 1% of exceeding the corresponding RF exposure limit. However, the safety study presented here was only a set of simulations examples meant to illustrate how the model may be used to follow the guidelines for active implants TS 10974:2018. Thus, the authors do not imply the safety of the VNS device for MRI given that, as discussed above, the safety study was only an example and would be otherwise incomplete.

We have introduced MARTIN, a detailed whole-body model for a male 29-month-old child, using the new automated segmentation tools for specific brain structures, as well as a manual segmentation performed by expert segmentors. Our model has been extensively validated, and the manuscript suggests how to perform MRI safety simulations on an AIMD by following the latest guidelines from the International Organization for Standardization. The model will be available on the Analogue Brain Imaging Laboratory (ABILAB) at the Athinoula A. Martinos Center for Biomedical Imaging website.

## Supporting information

**S1 File.**
(DOCX)

## Acknowledgments

Authors acknowledge to "Sim4Life by ZMT, www.zurichmeditech.com" for Science License and valuable feedback on AIMD Tier-3 analysis. The authors would also like to thank Jurriaan Peters for the discussion on the VNS implant, help with the implant positioning and trajectory.

**Disclaimer:** The mention of commercial products, their sources, or their use in connection with material reported herein was not to be construed as either an actual or implied endorsement of such products by the Department of Health and Human Services.

## Author Contributions

**Conceptualization:** Hongbae Jeong, Georgios Ntolkeras, Michel Alhilani, Lilla Zöllei, Kyoko Fujimoto, Michael H. Lev, P. Ellen Grant, Giorgio Bonmassar.

**Data curation:** Hongbae Jeong, Georgios Ntolkeras, Michel Alhilani, Seyed Reza Atefi, Lilla Zöllei, Michael H. Lev, P. Ellen Grant.

**Formal analysis:** Hongbae Jeong, Georgios Ntolkeras, Michel Alhilani, Seyed Reza Atefi, Lilla Zöllei, Kyoko Fujimoto, Ali Pourvaziri.

**Funding acquisition:** P. Ellen Grant, Giorgio Bonmassar.

**Methodology:** Hongbae Jeong, Georgios Ntolkeras, Michel Alhilani, Seyed Reza Atefi, Lilla Zöllei, Kyoko Fujimoto, Michael H. Lev, P. Ellen Grant, Giorgio Bonmassar.

**Project administration:** Giorgio Bonmassar.

**Resources:** P. Ellen Grant, Giorgio Bonmassar.

**Software:** Hongbae Jeong, Georgios Ntolkeras, Michel Alhilani, Seyed Reza Atefi, Lilla Zöllei.

**Supervision:** Michael H. Lev, P. Ellen Grant, Giorgio Bonmassar.

**Validation:** Georgios Ntolkeras, Michel Alhilani, Ali Pourvaziri, Michael H. Lev, P. Ellen Grant.

**Visualization:** Hongbae Jeong, Georgios Ntolkeras, Michel Alhilani, Seyed Reza Atefi, Lilla Zöllei.

**Writing – original draft:** Hongbae Jeong.

**Writing – review & editing:** Hongbae Jeong, Georgios Ntolkeras, Michel Alhilani, Seyed Reza Atefi, Lilla Zöllei, Kyoko Fujimoto, Ali Pourvaziri, Michael H. Lev, P. Ellen Grant, Giorgio Bonmassar.

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
