## [Decision Letter · Decision Letter 0]

18 Aug 2020

Pécs, Hungary

August 17, 2020

PONE-D-20-18921

Development, Validation, and Pilot MRI Safety Study of a High-Resolution, Open Source, Whole Body Pediatric Numerical Simulation Model

PLOS ONE

Dear Dr. Bonmassar,

Thank you for submitting your manuscript to PLOS ONE. After careful consideration, we feel that it has merit but does not fully meet PLOS ONE’s publication criteria as it currently stands. Therefore, we invite you to submit a revised version of the manuscript that addresses the points raised by the Reviewer, listed below.

We look forward to receiving your revised manuscript.

Kind regards,

Joseph Najbauer, Ph.D.

Academic Editor

PLOS ONE

Journal Requirements:

2. Please provide additional details regarding participant consent. In the ethics statement in the Methods and online submission information, please ensure that you have specified (1) whether consent was informed and (2) what type you obtained (for instance, written or verbal, and if verbal, how it was documented and witnessed). If the need for consent was waived by the ethics committee, please include this information.

Reviewers' comments:

Reviewer's Responses to Questions

**Comments to the Author**

1. Is the manuscript technically sound, and do the data support the conclusions?

Reviewer #1: Yes

2. Has the statistical analysis been performed appropriately and rigorously? 

Reviewer #1: Yes

3. Have the authors made all data underlying the findings in their manuscript fully available?

Reviewer #1: Yes

4. Is the manuscript presented in an intelligible fashion and written in standard English?

Reviewer #1: No

5. Review Comments to the Author

Reviewer #1: This manuscript details the development of a ~3 year old child body model for computational modeling, including definition of tissue properties such as dielectric values, and a pilot study of a vagus nerve stimulating device included in the model and simulated for MRI safety. The phantom development is remarkable and the body model is of great interest to the research community. However, several issues with the manuscript dampen enthusiasm for publishing in its present form. Most concerns may be addressed straightforwardly by editing details in the manuscript. A couple comments are raised on technical aspects of the study. I sincerely hope the authors revise this manuscript and resubmit it soon so their commendable work may be utilized by the research community.

Major comments:

1. The limitations of existing child body models are arguably overstated in the Introduction. The Roberta model from the Virtual Population indeed is anatomically accurate. The only issue is that Roberta is based on images from a 5 year old, whereas the present study prefers a 3 year old. If the manuscript is to detail at length the inadequacies of currently-available child body models, then this study should compare simulation results from MARTIN with results using Nina (which was morphed from Roberta). Typically, an important measure of a new tool for the research community is a rigorous comparison with the next best existing tool. There is a brief mention of comparison to the neonatal model reported by Malik et al 2015 [109] however that model does not appear to be included in Table 1, nor are details of that model discussed.

2. SSIM is a perception-based model, while alternative objective assessment measures, e.g., mean squared error, quantitatively describe absolute errors. SSIM is well suited for describing the effectiveness of an image compression algorithm, e.g., would a radiologist still perceive the relevant features of a compressed image versus the uncompressed image (based on contrast groups, as the authors state)? With so much care directed to the precise tissue parameters, e.g., T1, T2, dielectric properties, etc., why base the analysis of the new body phantom with the Bloch simulator on SSIM measures instead of MSE? Is the goal of the Bloch simulator aspect of the study to determine if the body model will produce images that are perceived to be similar to actual MRI, or is the goal to gauge how well the body model and appropriate tissue properties are a theoretical match to actual MRI?

Minor comments:

3. In the Materials and Methods subsection on The Validation, calculation of tissue compartment density is discussed. Then, in the Dielectric properties subsection there is a rigorous discussion of the change of dielectric properties with age. Does the literature suggest any variations in tissue density based on age? If so, perhaps this should also be considered in this study.

4. The Pilot simulation study is unclear regarding the type of solver in Sim4Life that was used. Was it FEM or FDTD? It seems it was FEM, since much attention is paid to the surface mesh extraction and unstructured tetrahedral mesh surfaces, however it remains unclear.

5. The Introduction motivates the study by reviewing the lack of validated child body models based on child image sets, as well as the safety questions surrounding MRI of children with VNS. Since it is mentioned PubMed does not presently include studies focusing on MRI safety of children with VNS, it would be beneficial to include a brief survey of existing MRI safety studies for adults with VNS. The one reference in the manuscript is Shellock et al. [43], which was published 16 years ago. I would suggest adding more recent literature to the Introduction to motivate this child body model and simulation project. Examples include de Jonge et al. 2014 (doi: 10.1111/epi.12774), which included 11 patients between 5-12 years old, Gorny et al. 2010 (doi: 10.1002/jmri.22037), and Rösch et al. 2015 (doi: 10.1016/j.eplepsyres.2014.11.010).

6. In the Introduction, Wolf et al. [34] is cited to substantiate the argument that low-quality models can lead to SAR underestimation. Wolf referred to the original Virtual Family models not including CSF in the brain, thus removing the highest conductivity tissue from the head; CSF was added in later versions. This does not exactly substantiate the argument that a different sized organ, e.g., the heart, will lead to underestimation, even if that argument is correct.

7. The authors discuss the advantage of this study is the use of in vivo medical images of a 29-month-old child and the age appropriateness of the model tied to the subject's age. However, given the CT data were acquired when the boy was 24 months old, the gap and required deformation seems to undercut part of the stated motivation for this study (not morphing or adjustments). On a second note, given 29 months is closer to 2 years compared to 3 years, perhaps the model should not be described as a three year old child.

8. The reference list needs to be carefully reviewed for formatting errors. [6] and [83] incorrectly abbreviate organization or document names as if they were a given name and surname. [23] misspells the first author's surname. [42] does not provide enough information to be useful. [81] does not include a publication year.

9. Table 1 indeed lists "models currently available" however they are all not necessarily "state-of-art" unless they were released in recent years. If some are older, I suggest removing "state-of-art". However, perhaps the table should include the year each model was released, as that provides additional valuable information.

10. Table 1 states the Virtual Population models Charlie, Nina, and Roberta are not freely accessible to the research community. In fact, V1.x of those models are available to the academic community for a nominal shipping and handling charge for the data DVD.

11. Table 1 states "Information not available" for two of the phantoms. Did the authors inquire as to the availability of these phantoms? I know this is not a journalism manuscript, but still some may consider it worthwhile to ask to ascertain details before reporting there is no available information.

12. To improve readability, the manuscript would benefit from careful editing by a native English speaker. Some changes suggested by this reviewer follow:

- Intro: change "computational with human body modeling" to "computational modeling with the human body"

- Intro: "electromagnetism (2)" to "electromagnetics (2)" (note how other areas listed tend to end with -ics)

- Intro: "computational tomography" should be "computed tomography"

- Intro: "segmented into 3D whole-body voxel model" should be "segmented into a 3D ..."

- Intro: Virtual Family should have capitalized first letters

- Intro: "various ages ranging between an eights-weeks and an 80-years-old" should be "various ages ranging between eight weeks and 80 years" (grammatically, please note the age should be hyphenated if used as a compound modifier, but otherwise not hyphenated)

- Intro: "using images from a whole body in-vivo subject." should be "using images from whole body in-vivo subjects." if you are referring to all the Virtual Family and not just one model. If you are only referring to one model in the Virtual Family, then it is correct as written but the preceding sentence should be modified as it refers to multiple Virtual Family models.

- Intro: "neck model is MIDA" should be "neck models is MIDA"

- Intro: "do not grow symmetrically" -- do the authors mean "proportionally"? "Symmetrically" implies left-right symmetry as opposed to different organs growing proportionally.

- Intro: "it is a 5-year-old model" -- is this referring to the age of the model (present writing implies this) or referring to the age of the child that is modeled? If the latter, suggest "is a model of a 5 year old".

- Intro: "66 tissue segmentation" to "66 tissues segmented"

- Intro: "Having both more tissues are visible for accurate segmentation" -- please rewrite, the message is confusing as written.

- Intro: "includes 86 tissue segmentation" to "includes 86 segmented tissues"

- Methods: suggest standardizing to the vendor and/or scanner model being all caps or not all caps, e.g., SIEMENS TrioTim, Siemens SOMATOM, Siemens Healthinears, SIEMENS Healthinears.

- Methods: regarding the above sentences, note the typos "Healthinears" should be "Healthineers"

- Methods: suggest describing 3D Slicer as "an open-source software platform" at the first mention of 3D Slicer in the Data co-registration subsection.

- Methods: "not covered by a CT scan" -- suggest an alternative verb to covered

- Methods: "P.E.G, and M.H.L." to "P.E.G. and M.H.L." (note this typo occurs twice in this section)

- Methods: there is no need to repeat the phrase "two experienced, sub-specialty certified neuro-radiologists" or similar phrases many times. After first stating this, the authors can simply state "certified neuro-radiologists" or simply "neuro-radiologists". Variations of this lengthy phrase also appear later.

- Methods: "Sim4life" to "Sim4Life"

- Methods: "extracted, and curvature smoothing was done" to "extracted, curvature smoothing was done"

- Regarding the above suggestion, the verb "done" is used many times throughout the manuscript. More precise verbs would improve the readability.

- Methods: "Technical Specification (ISO/TS) 10974 Technical Specification" seems to have unnecessary repeated words

- Methods: Mentioning Sim4Life as a tool used in the study does not require a reference to the product [65] unless the authors are referencing a specific detail in the documentation.

- Results: "result was put together" -- suggest rewording

- Results: "4.2 % and 3.3 larger" to "4.2% and 3.3% larger"

- Results: "of they the thymus" to "of the thymus"

- Results: "were found high in the bone" to "were found in the bone"

- Results: "of a MARTIN of a pilot use case of with and without" to "of a pilot use case of MARTIN with and without"

- Discussion: "inter-subject data" should be "intra-subject data", since it was the same subject, at least that is how it reads now

- Discussion: "opportunity for ones to separate" to "opportunity for one to separate"

- Discussion: "other regions of and other limbs" -- remove "of"

- Discussion: "reported in this study with MARIN" -- typo for MARTIN

6. PLOS authors have the option to publish the peer review history of their article (what does this mean?). If published, this will include your full peer review and any attached files.

Reviewer #1: No

---

## [Author Response · Author response to Decision Letter 0]

7 Oct 2020

Please see Response to the Reviewer.

---

## [Decision Letter · Decision Letter 1]

20 Oct 2020

Pécs, Hungary

October 19, 2020

Development, Validation, and Pilot MRI Safety Study of a High-Resolution, Open Source, Whole Body Pediatric Numerical Simulation Model

PONE-D-20-18921R1

Dear Dr. Bonmassar,

We’re pleased to inform you that your manuscript (R1 version) has been judged scientifically suitable for publication and will be formally accepted for publication once it meets all outstanding technical requirements.

PLEASE SEE COMMENTS BY THE REVIEWER AND ACADEMIC EDITOR BELOW.

Kind regards,

Joseph Najbauer, Ph.D.

Academic Editor

PLOS ONE

Additional Editor Comments:

5 Center for Devices and Radiological Health, U.S. Food and Drug Administrator, Silver Spring, MD, United States

Please change to

5 Center for Devices and Radiological Health, U.S. Food and Drug Administration, Silver Spring, MD, United States

Reviewers' comments:

Reviewer's Responses to Questions

**Comments to the Author**

1. If the authors have adequately addressed your comments raised in a previous round of review and you feel that this manuscript is now acceptable for publication, you may indicate that here to bypass the “Comments to the Author” section, enter your conflict of interest statement in the “Confidential to Editor” section, and submit your "Accept" recommendation.

Reviewer #1: (No Response)

2. Is the manuscript technically sound, and do the data support the conclusions?

Reviewer #1: Yes

3. Has the statistical analysis been performed appropriately and rigorously? 

Reviewer #1: Yes

4. Have the authors made all data underlying the findings in their manuscript fully available?

Reviewer #1: Yes

5. Is the manuscript presented in an intelligible fashion and written in standard English?

Reviewer #1: Yes

6. Review Comments to the Author

Reviewer #1: I commend the authors for the thorough response to previous comments, including the comment for which the reviewer offered outdated information (e.g., IT'IS offering Virtual Population models freely to the research community). This reviewer thanks the authors for the clear and direct responses to previous comments.

The newly added comparison between MARTIN and Nina strengthens the novelty and utility of the authors' work, as does the MSE analysis for validation.

The Reference List entry for [23] still needs to correct the authors' surnames. Current: "PA H, F DG, C B, Neufeld E LB, MC G, D P, et al." Correct: "Hasgall P, Di Gennaro F, Baumgartner C, Neufeld E, Lloyd B, Gosselin M, et al."

7. PLOS authors have the option to publish the peer review history of their article (what does this mean?). If published, this will include your full peer review and any attached files.

Reviewer #1: **Yes: **Joseph V Rispoli

---

## [Editor Report · Acceptance letter]

22 Dec 2020

PONE-D-20-18921R1 

Development, Validation, and Pilot MRI Safety Study of a High-Resolution, Open Source, Whole Body Pediatric Numerical Simulation Model

Dear Dr. Bonmassar:

I'm pleased to inform you that your manuscript has been deemed suitable for publication in PLOS ONE. Congratulations! Your manuscript is now with our production department. 

Kind regards, 

on behalf of

Dr. Joseph Najbauer 

Academic Editor

PLOS ONE